# Atypical plant homeodomain of UBR7 functions as an H2BK120Ub ligase and breast tumor suppressor

Santanu Adhikary[1,2], Deepavali Chakravarti[3,4], Christopher Terranova[3], Isha Sengupta[1], Mayinuer Maitituoheti[3], Anirban Dasgupta[2], Dushyant Kumar Srivastava[2], Junsheng Ma[5], Ayush T. Raman[3], Emily Tarco[6], Aysegul A. Sahin[7], Roland Bassett[5], Fei Yang[6], Coya Tapia[7], Siddhartha Roy[2], Kunal Rai[3] & Chandrima Das[1]

The roles of Plant Homeodomain (PHD) fingers in catalysis of histone modifications are unknown. We demonstrated that the PHD finger of Ubiquitin Protein Ligase E3 Component N-Recognin7 (UBR7) harbors E3 ubiquitin ligase activity toward monoubiquitination of histone H2B at lysine120 (H2BK120Ub). Purified PHD finger or full-length UBR7 monoubiquitinated H2BK120 in vitro, and loss of UBR7 drastically reduced H2BK120Ub genome-wide binding sites in MCF10A cells. Low UBR7 expression was correlated with occurrence of triple-negative breast cancer and metastatic tumors. Consistently, UBR7 knockdown enhanced the invasiveness, induced epithelial-to-mesenchymal transition and promoted metastasis. Conversely, ectopic expression of UBR7 restored these cellular phenotypes and reduced tumor growth. Mechanistically, UBR7 loss reduced H2BK120Ub levels on cell adhesion genes, including CDH4, and upregulated the Wnt/β-Catenin signaling pathway. CDH4 overexpression could partially revert UBR7-dependent cellular phenotypes. Collectively, our results established UBR7 as a histone H2B monoubiquitin ligase that suppresses tumorigenesis and metastasis of triple-negative breast cancer.

[1] Biophysics and Structural Genomics Division, Saha Institute of Nuclear Physics, 1/AF Bidhan Nagar, Kolkata 700064, India. [2] Structural Biology and Bio-Informatics Division, CSIR-Indian Institute of Chemical Biology, 4 Raja S.C. Mullick Road, Kolkata 700032, India. [3] Department of Genomic Medicine, The University of Texas MD Anderson Cancer Center, Houston, TX 77030, USA. [4] Department of Cancer Biology, The University of Texas MD Anderson Cancer Center, Houston, TX 77030, USA. [5] Department of Biostatistics, The University of Texas MD Anderson Cancer Center, Houston, TX 77030, USA. [6] Department of Translational Molecular Pathology and Department of Investigational Cancer Therapeutics, The University of Texas MD Anderson Cancer Center, Houston, TX 77030, USA. [7] Department of Pathology, The University of Texas MD Anderson Cancer Center, Houston, TX 77030, USA. These authors contributed equally: Santanu Adhikary, Deepavali Chakravarti, Christopher Terranova. Correspondence and requests for materials should be addressed to S.R. (email: roysiddhartha@iicb.res.in) or to K.R. (email: krai@mdanderson.org) or to C.D. (email: chandrima.das@saha.ac.in)

Breast cancer is the most common cause of cancer mortality in female individuals. The heterogeneity of the disease poses immense challenges in deciphering therapeutic strategies[1]. The hormone receptor-negative or triple-negative subtype has the worst prognoses due to the lack of targeted therapies[2,3]. Although accumulation of genetic defects has been involved in the development of oncogenesis, epigenetic abnormalities play a significant role in the initiation, progression, and metastasis of the disease[4]. Specifically, epithelial-to-mesenchymal transition (EMT), which preludes the onset of metastasis[5,6], is thought to be driven by epigenetic alterations[7,8]. Primarily, histone modifications, which include methylation, acetylation, and ubiquitination, play crucial roles in maintaining homeostasis, failure of which may lead to disease initiation or progression[9]. Importantly, breast cancer cases with worse prognoses have lower levels of H3K18Ac, H4K12Ac, H3K4Me2, H4K20Me3, and H4R3Me2 marks[9,10].

Monoubiquitination at lysines on histones H2A and H2B have an antagonistic relationship in oncogenesis[11–14]. In contrast with polyubiquitination, which marks the protein for its proteasome-mediated degradation, monoubiquitination of histone H2B plays key roles in transcription memory and elongation, DNA damage response, viral infection, stem cell differentiation, and oncogenesis[15,16]. The E3 ligases for H2B monoubiquitination, RNF20 and RNF40, are reported to act as potent tumor suppressors, regulate DNA double-stranded break repair, and modulate stem cell differentiation[14,17,18].

The ubiquitin protein ligase E3 component N-recognin (UBR) family of mammalian E3 ligases containing seven members (UBR1–UBR7) is characterized by a 70-residue zinc finger-type UBR-box domain, which is essential for recognition of the N-degrons[19–21]. Despite harboring a UBR-box, UBR3, UBR6, and UBR7 do not bind to N-degrons. Although the members of the UBR family of proteins are generally heterogeneous in size and sequence, they harbor specific signatures unique to ubiquitin ligases or a substrate-recognition subunit of the E3 complex like the RING/HECT (really interesting new gene/homologous to the E6AP carboxyl terminus) domain or F-box[21,22]. A RING domain is present in UBR1, UBR2, and UBR3, a HECT domain is present in UBR5, and an F-box is present in UBR6. Of note, UBR7 has evolved with a plant homeodomain (PHD) finger, which is a putative chromatin-binding module, not present in any other UBR family proteins. Although the PHD finger is well characterized as a reader of methylated, acetylated, or unmodified histones[23], its role in enzymatic catalysis is not known. Furthermore, little is known about the role of UBR7 in carcinogenesis. In the present study, we demonstrated that the UBR7-PHD finger is an H2BK120 monoubiquitin ligase and a tumor suppressor in triple-negative breast cancer cases.

## Results

**UBR7-PHD monoubiquitinates histone H2B lysine 120 in vitro**. UBR7, a protein with an unknown function, contains a UBR-box domain, which is essential for the recognition of N-degrons[20–22], and a PHD finger (Fig. 1a), which is highly conserved across species (Supplementary Fig. 1a). Although the sequence alignment of the UBR7-PHD finger (which is stabilized by zinc ion coordination in a cross-braced topology) with other well-characterized H3K4Me3 or H3K4Me0 binders exhibited several conserved residues, it displayed weak interaction with trimethylated histone peptides (Supplementary Fig. 1b–f). Although full-length UBR7 protein interacted with all recombinant histones in vitro, the PHD finger preferentially interacted with recombinant histone H2B (Fig. 1b) and could also immunoprecipitate them from MCF10A cells (Supplementary Fig. 1g). Sequence alignment of the UBR7-PHD with other classical RING

finger-E3 ubiquitin ligases indicated that zinc-coordinating His163 and His166 are unique in contrast to the other RING fingers (Supplementary Fig. 1b). Site-directed mutagenesis of H163S/H166S of the UBR7-PHD did not significantly compromise its association with histone H2B at various levels of organization (Fig. 1c). We observed similar results during immunoprecipitation assays from HEK293T cells (Fig. 1d). However, mutation of lysine 120 to arginine (K120R) in histone H2B abrogated its binding preference for UBR7 as observed through immunoprecipitation assays (Fig. 1e). Based on the zinc coordination fold similarity between the RING and the PHD finger, we hypothesized that UBR-PHD function as an E3 ubiquitin ligase for histone H2B substrate. Purified recombinant UBR7 full-length wild-type protein (UBR7-WT), full-length H163S/H166S catalytic-mutant (UBR7-CM), or individual domains (UBR or PHD) were incubated in the presence of an E1 ubiquitin-activating enzyme, an E2 ubiquitin-conjugating enzyme (UbcH6), ATP, inorganic pyrophosphatase, and biotin-tagged ubiquitin. UBR7-WT and the PHD finger alone could monoubiquitinate purified H2B, H2A/H2B dimer, core histone octamers, and purified nucleosomes (Fig. 1f, g and Supplementary Fig. 1h–k), in contrast with the other E3 ubiquitin ligases, which usually act in complex[24–26]. On the other hand, UBR7-CM failed to promote H2B ubiquitination (Fig. 1f, g and Supplementary Fig. 1j–o). Thus, our results demonstrated the E3 ubiquitin ligase function of UBR7 toward monoubiquitination of histone H2B in vitro.

**UBR7 regulates H2BK120Ub levels ex vivo**. Next, we sought to determine whether UBR7 regulates H2BK120Ub in mammalian cells. We observed lower UBR7 levels in human and murine breast cancer cells than in their "normal" counterparts. For example, UBR7 messenger RNA (mRNA) transcript and protein levels were lower in MCF7, T47D, MDA-MB-231, and MDA-MB-468 cells than in MCF10A and MCF12A cells (Fig. 2a and Supplementary Fig. 2a). We observed similar patterns in 21PT and 21MT2 compared to 16N (human) and 4T07 compared to 4T1 (murine) cells (Supplementary Fig. 2b-e). Genetic depletion of UBR7 by two different short hairpin RNAs (shRNAs) in MCF10A and MCF12A cells led to a dramatic reduction in global H2BK120Ub levels (Fig. 2b and Supplementary Fig. 2f, g). Importantly, the reduction caused by UBR7-shRNA was rescued by UBR7-WT, but not by UBR7-CM (Fig. 2c and Supplementary Fig. 2h). Consistently, UBR7-WT, but not UBR7-CM, increased H2BK120Ub levels in the MDA-MB-231 and MDA-MB-468 breast cancer cells (Fig. 2d, e and Supplementary Fig. 2h).

To examine changes in H2BK120Ub genome wide, we performed chromatin immunoprecipitation sequencing (ChIP-seq) of H2BK120Ub in control and UBR7-knockdown MCF10A cells. UBR7-knockdown cells had a drastically lower number of H2BK120Ub binding sites (1079) compared to control cells (8401) (Fig. 2f and Supplementary Data 1), which was verified by ChIP-quantitative polymerase chain reaction (qPCR) (Fig. 2g). Similarly, the intensity of H2BK120Ub enrichment was drastically reduced as demonstrated by average density plots and verified by ChIP-qPCR for selected genes (Fig. 2h, i and Supplementary Fig. 2i, j).

To analyze the impact of UBR7-mediated H2BK120Ub on the chromatin landscape, we also performed ChIP-seq for histone modifications H3K79Me2 (transcription), H3K4Me3 (promoters), H3K4Me1 (enhancers), H3K27Ac (active enhancers), H3K27Me3 (polycomb-repressed), and H3K9Me3 (heterochromatin)[27]. Consistent with prior reports, we noted a loss of H3K79Me2 on H2BK120Ub gene targets upon UBR7 knockdown (Fig. 2j). Identification of chromatin state transitions between control and UBR7-depleted cells using the ChromHMM

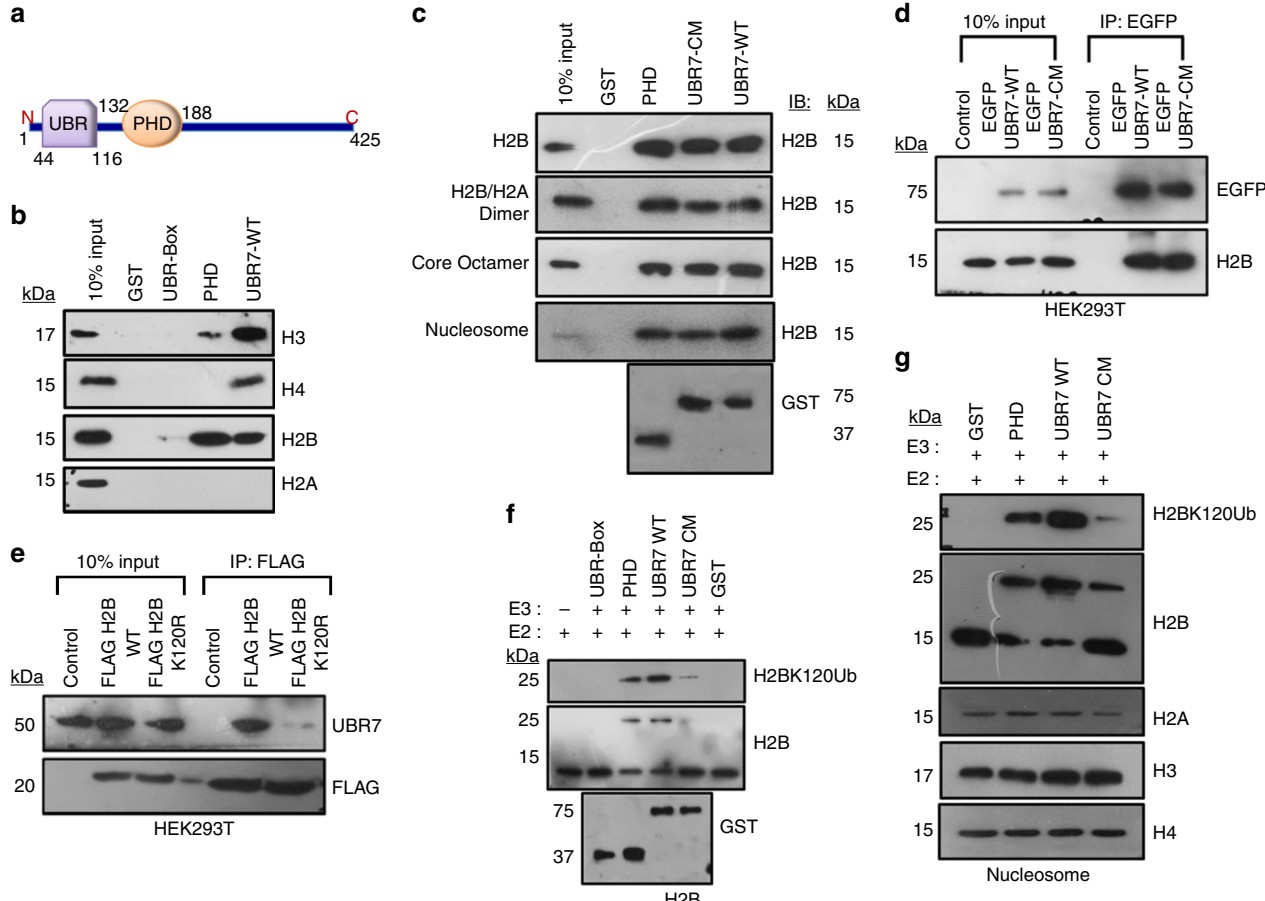

**Fig. 1** UBR7 is a histone H2B ubiquitin ligase. **a** Schematic of the domain organization of UBR7. **b** Immunoblots showing the in vitro interaction of purified UBR7 with recombinant histones H3, H4, H2B, and H2A. **c** Interaction of UBR7 full-length wild-type (WT) or catalytic-mutant (CM) with recombinant H2B, H2A/H2B dimer, core octamer, or purified nucleosomes from HeLa cells. **d** Ex vivo interaction of H2B with UBR7 WT or CM in HEK293T cells. **e** Ex vivo interaction of UBR7 with H2B WT or H2B mutant (K120R) in HEK293T cells. **f, g** In vitro ubiquitination assay with recombinant H2B (**f**), or purified nucleosomes from HeLa cells (**g**)

algorithm in a 10-state model (Fig. 2k; see Methods) indicated that H2BK120Ub was primarily present in conjunction with H3K79Me2. Also, the most potent transitions were from highly transcribed states in control cells to low/non-transcribed states in UBR7-depleted cells (States 1 to 5, States 2 to 4 or 1, States 3 to 5, and States 4 to 6) (Fig. 2k, l). Overall, these experiments identified UBR7 as an E3 ubiquitin ligase in vivo and demonstrated the importance of UBR7 in maintaining specific chromatin patterns in cells.

Next, we defined the relative impact of UBR7 and other two known H2BK120Ub E3 ligases, RNF20 and RNF40, on its levels and genomic distribution. Knockdown of all the three E3s, UBR7, RNF20, and RNF40, separately in MCF10A cells led to significant reduction of total H2BK120Ub levels (Supplementary Fig. 3a). Direct comparison between ChIP-seq profiles for H2BK120Ub showed drastic losses at 10,919 sites in UBR7-deficient, 11,005 sites in RNF20-deficient, and 11,069 sites in RNF40-deficient cells (Supplementary Fig. 3b–g). Sites that lose H2BK120Ub upon individual knockdown of each of these enzymes significantly overlapped (Supplementary Fig. 3h–m), suggesting cooperative interaction between UBR7, RNF20, and RNF40 and requirement of all three proteins for maintenance of H2BK120Ub levels in the cell.

**Low UBR7 is associated with triple-negative breast cancer.** Next, to further explore UBR7's biological function, we examined

its expression in The Cancer Genome Atlas Group (TCGA) breast cancer mRNA expression data[28]. This analysis revealed reduced UBR7 expression in triple-negative and basal-like breast tumors (Fig. 3a), which was confirmed in an independent cohort of breast cancer tissue microarrays (TMAs) containing 371 breast tumors using a UBR7-specific antibody (Supplementary Fig. 4a and Supplementary Data 2). We found that lower levels of UBR7 correlated significantly with triple-negative status (Fig. 3b) and individual estrogen receptor (ER) status and progesterone receptor (PR) status (Fig. 3c, d and Supplementary Data 2). Also, metastatic intraductal carcinomas harbored considerably lower levels of UBR7 than did primary tumors (Fig. 3e, f). Furthermore, UBR7 was associated with a better metastasis-free survival rate in the aggressive basal-type breast cancer (mesenchymal subtype tumors without endocrine therapy) (Supplementary Fig. 4b).

**UBR7 acts as tumor metastasis suppressor gene.** A series of loss-of-function and gain-of-function experiments established the role of UBR7 as a tumor and metastasis suppressor gene. Reduction of UBR7 in MCF10A and MCF12A cells increased two-dimensional (2D) proliferation drastically (Fig. 3g and Supplementary Fig. 4c), as well as expression of the proliferation markers Ki-67, PCNA, and MCM2[29] (Fig. 3h–j and Supplementary Fig. 4d). Overexpression of UBR7-WT and not UBR7-CM reversed the increased proliferation observed in UBR7-knockdown MCF10A cells (UBR7-sh1) (Fig. 3k) and the

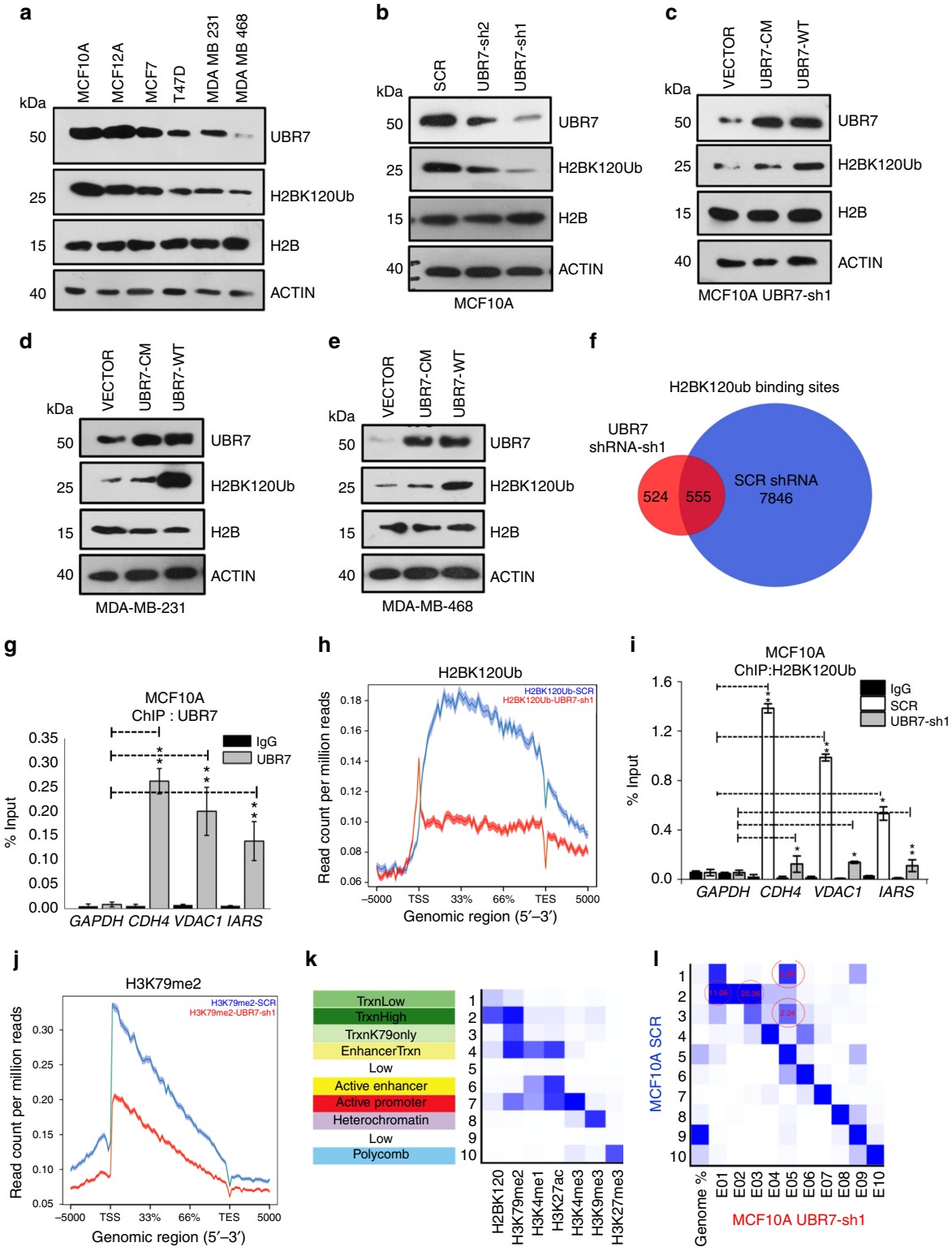

expression of associated proliferation markers (Fig. 3l and Supplementary Fig. 4e, f). Similarly, overexpression of UBR7-WT, but not UBR7-CM, in MDA-MB-231 and MDA-MB-468, two basal-like breast cancer cells that express UBR7 at low levels, dramatically reduced 2D proliferation (Supplementary Fig. 4g, h). Next, we examined three-dimensional (3D) soft agar colony growth of UBR7-depleted MCF10A cells in vitro and found

substantially higher numbers of colonies as well as bigger colonies in UBR7-depleted cells than in control (Fig. 4a, b). Importantly, the impact of wild-type derivatives in the 3D growth assay in MDA-MB-231 cells was abrogated by UBR7-CM (Fig. 4c, d). Overexpression of UBR7-WT and not UBR7-CM abrogated mammary fat pad tumor formation in vivo (Fig. 4e, f). Additionally, immunohistochemical analysis of the proliferation

**Fig. 2** UBR7 is downregulated in invasive breast cancer cells. **a** Immunoblots of MCF10A, MCF7, T47D, MDA-MB-231, and MDA-MB-468 cells to monitor expression of UBR7, H2BK120Ub, and H2B. ACTIN was used as a loading control. **b** Immunoblots for UBR7, H2BK120Ub, H2B, and ACTIN (loading control) in MCF10A cells expressing scrambled (SCR), UBR7-sh1, or UBR7-sh2 shRNA. **c–e** Immunoblots for UBR7, H2BK120Ub, H2B, and ACTIN (loading control) in MCF10A UBR7-sh1 (**c**), MDA-MB-231 (**d**), and MDA-MB-468 (**e**) cells expressing wild-type (UBR7-WT) and catalytic-mutant (UBR7-CM) UBR7. ACTIN was used as a loading control. **f** Venn diagram showing overlap of total H2BK120Ub binding sites in Control (SCR) and UBR7-sh1-expressing MCF10A cells. **g** Bar plot for quantitative PCR (qPCR) enrichment of UBR7 chromatin immunoprecipitation (ChIP) in MCF10A cells for selected genes. *GAPDH* was used as a negative control. **h** Average genebody density plot for H2BK120Ub binding sites in Control (SCR) and UBR7-sh1-expressing MCF10A cells. **i** Bar plot for qPCR enrichment of H2BK120Ub ChIP in MCF10A cells expressing SCR or UBR7-sh1. *GAPDH* was used as a negative control. **j** Average genebody density plot for H3K79me2 binding sites in Control (SCR) and UBR7-sh1-expressing MCF10A cells. **k** Emission parameter for a 10-state chromatin state model called by the default parameters of ChromHMM. States in the left column were annotated based on their closeness to the nearest transcription start sites (TSS) and nature of constituent marks. **l** Overlap enrichment analysis displaying chromatin state transitions between MCF10A control (SCR) cells (*Y*-axis) and MCF10A UBR7-sh1 cells (*X*-axis). The most significant state transitions include losses of H2BK120Ub/H3K79me2 low (States 1 to 5), H2BK120Ub/H3K79me2 high (States 2 to 1 or 3) and H3K79me2 only (States 3 to 5), which are highlighted by red circles. In **g**, **i**, error bars indicate standard deviation (s.d.); *n* = 3 technical replicates of a representative experiment (out of three experiments). *P* values were calculated using two-tailed *t* tests. *$P < 0.05$; **$P < 0.001$

---

**Table 1 Five chosen GO terms from DAVID on differentially expressed genes in UBR7-knockdown MCF10A cells (FDR <0.01; FC >2)**

| GO TERM | FDR |
| --- | --- |
| GO:0006414~ translational elongation | 2.0E−43 |
| GO:0007049~ cell cycle | 5.3E−16 |
| GO:0007155~ cell adhesion | 1.3E−05 |
| GO:0031497~ chromatin assembly | 5.8E−05 |
| GO:0016126~ sterol biosynthesis process | 1.5E−03 |

*DAVID* Database for Annotation, Visualization and Integrated Discovery, *GO* gene ontology, *FC* fold change, *FDR* false discovery rate

---

marker Ki-67, tumors derived from mice, confirmed the anti-proliferative role of UBR7-WT, but not UBR7-CM (Fig. 4g). These results supported a tumor-suppressive role for UBR7 in breast cancer.

Consistent with a metastasis-suppressive role for UBR7 and its lower levels in metastatic tumors, UBR7-WT overexpressing MDA-MB-231 cells, but not UBR7-CM-overexpressing MDA-MB-231 cells, were unable to seed to the lung upon intravenous injection, whereas control cells formed overt lung metastases (Fig. 4h). Consistently, UBR7 loss in MCF10A cells enhanced invasion in a Matrigel chamber (Fig. 4i, j) and migration in a scratch assay (Fig. 4k, l and Supplementary Fig. 5a, b). Importantly, these phenotypes were rescued by overexpression of UBR7-WT, but not UBR7-CM, in MCF10A UBR7-sh1 cells (Fig. 4m–p), as well as in MDA-MB-231 and MDA-MB-468 cells (Supplementary Fig. 5c–h). Overall, these results establish the tumor- and metastasis-suppressive functions of UBR7 in triple-negative breast cancer.

**UBR7 suppresses EMT.** To gain insight into the molecular mechanism of these observations, we performed transcriptomic profiling of MCF10A cells harboring control and UBR7-specific shRNA using RNA-sequencing (RNA-seq) and identified 2348 up-regulated and 2576 downregulated genes (Fig. 5a, Supplementary Fig. 6a, b and Supplementary Data 3). We verified a subset of these genes using individual qPCR experiments (Supplementary Fig. 6c, d). Consistent with the cellular phenotypes, the misregulated genes exhibited enrichment in Cadherin and invasive breast cancer signatures (Fig. 5b, Supplementary Data 4 and Table 1). Importantly, we noted that UBR7-low (shRNA harboring) cells had characteristics of epithelial-to-mesenchymal

transition (EMT), a cellular process typically associated with breast cancer metastasis[5], as judged by loss of expression of epithelial markers (*CDH1*, *CLDN1*, *CLDN7*, and *CYTK18*) and gain of expression of mesenchymal markers (*CDH2*, *ZEB1*, *SNAI1*, *SNAI2*, *TWIST*, *VIM*)[6,30] in qPCR, western blot, and immunofluorescent analyses (Fig. 5c–f and Supplementary Fig. 7a–c). Overexpression of UBR7-WT in MCF10A UBR7-shRNA harboring cells (UBR7-sh1) rescued the expression of these markers, whereas UBR7-CM did not (Fig. 5g–i). Similarly, overexpression of UBR7-WT, but not UBR7-CM, reduced the expression of mesenchymal markers and induced the expression of epithelial markers in MDA-MB-231 and MDA-MB-468 cells (Fig. 5j, k and Supplementary Fig. 7d–g). Therefore, our results indicated that loss of UBR7 promotes EMT, a phenomenon thought to precede metastasis.

**UBR7 suppresses EMT through activation of CDH4.** To define the UBR7 gene targets that may drive the observed phenotypes, we overlapped sites that harbored loss of H2BK120Ub in UBR7-knockdown cells using ChIP-seq (7846, $p < 1e-8$) with differentially expressed genes (4924; $p < 0.01$; fold change >1.5) and found that H2BK120Ub targeted 318 downregulated and 117 up-regulated genes (Fig. 6a, Supplementary Fig. 8a, b, and Supplementary Data 5). Several such genes were enriched in cell–cell adhesion processes (Supplementary Data 6) including cadherins such as *CDH4* and *CDH13* (Fig. 6b and Supplementary Fig. 8c–e). *CDH4* (or R-cadherin) is suggested to play important roles in suppressing invasion of basal-type breast cancer[31]. Furthermore, *CDH4* expression exhibited consistent patterns to UBR7 in matched "normal" and "malignant" breast cancer cell lines (Fig. 6c and Supplementary Fig. 8f). Importantly, we detected UBR7 occupancy in *CDH4* locus in control MCF10A cells using ChIP-qPCR as well as rescue of H2BK120Ub levels in MCF10A UBR7-sh1, MDA-MB-231, and MDA-MB-468 cells by overexpression of UBR7-WT, but not UBR7-CM (Figs 2g, 6d–f). Next, we sought to determine whether CDH4 was in part responsible for anti-invasive phenotypes of UBR7. Overexpression of CDH4 in MCF10A and MCF12A UBR7-sh1 cells drastically reduced cellular invasion, migration, proliferation, and suppressing EMT, and established an epistatic relationship between UBR7 and CDH4 (Fig. 6g–m and Supplementary Figs 8g–j, 9a,b). Similarly, CDH4 overexpression drastically reduced the invasive properties of both MDA-MB-231 and MDA-MB-468 cells (Supplementary Fig. 9c–i). Overall, our results indicated that functional loss of UBR7 can be restored by cell–cell adhesion genes like *CDH4*.

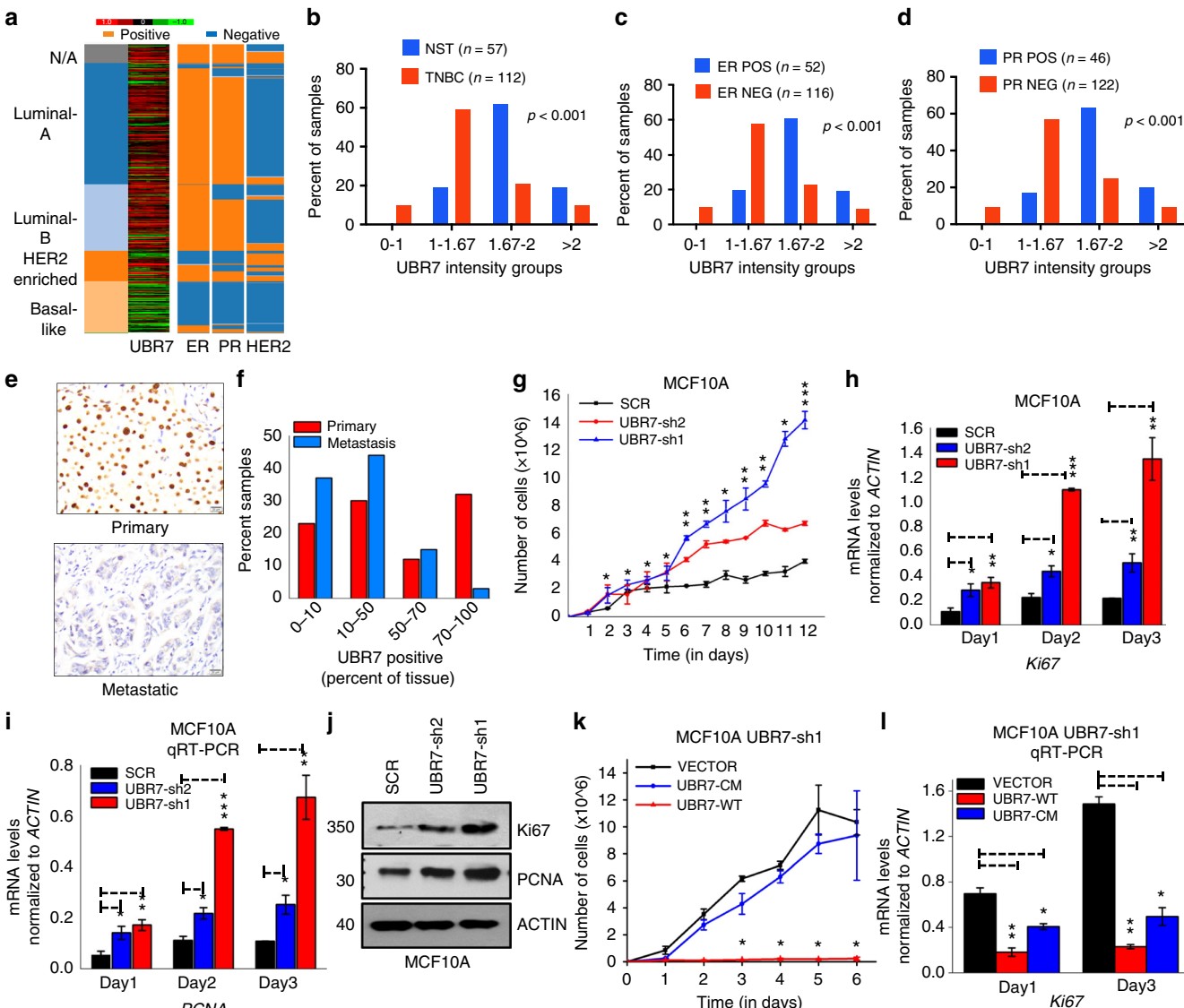

**Fig. 3** UBR7 inhibits breast cancer cell proliferation. **a** Heatmap showing that *UBR7* expression is downregulated in triple-negative or basal-type breast cancer (analysis of data from TCGA database). Green and red indicates down- and up-regulated, respectively. **b–d** Immunohistochemical analysis representing low UBR7 expression analyzed in triple-negative (**b**), estrogen receptor (ER)-negative (**c**), and progesterone receptor (PR)-negative (**d**) breast tumors. **e, f** Representative images of primary and metastatic invasive breast carcinoma showing UBR7 expression as analyzed immunohistochemically. Scale bar indicates 20 μm. **g** Proliferation of MCF10A cells expressing scrambled (SCR) or UBR7 (UBR7-sh1/UBR7-sh2) short hairpin RNAs (shRNAs). **h, i** Quantitative real-time PCR (qRT-PCR) analysis of *Ki-67* (**h**) and *PCNA* (**i**) from cultured MCF10A cells expressing SCR or UBR7 (UBR7-sh1/UBR7-sh2) shRNAs in a time-dependent manner. **j** Immunoblots of MCF10A cells expressing SCR or UBR7-shRNAs to monitor expression of Ki-67 and PCNA. ACTIN was used as a loading control. **k** Proliferation of cultured MCF10A UBR7-sh1 cells expressing a vector (VECTOR), WT (UBR7-WT), or catalytic-mutant (UBR7-CM). **l** qRT-PCR analysis of *Ki-67* in MCF10A UBR7-sh1 cells expressing vector (VECTOR), wild-type (UBR7-WT), or CM (UBR7-CM) in a time-dependent manner. In all panels, error bars indicate standard deviation (s.d.); $n = 3$ technical replicates of a representative experiment (out of three experiments). *P* values were calculated using two-tailed *t* tests. *$P < 0.05$; **$P < 0.001$; ***$P < 0.0001$

**UBR7 silencing activates Wnt/β-catenin signaling**. The Wnt/β-catenin signaling pathway is deregulated in several cancers, including breast cancer[32]. It promotes tumor initiation, maintenance and metastasis[32]. We observed activation of the canonical Wnt/β-catenin signaling cascade upon loss of UBR7 as evident from the upregulation of key positive regulators including WNT3A, FZD2/3, LRP5/6, ROR2, and DSH2, and down-regulation of negative regulators such as DKK1 (Fig. 7a, b and Supplementary Fig. 9j). Activation of the signaling pathway was further confirmed by nuclear localization of β-catenin upon loss of UBR7 (Fig. 7c). This could serve as a mechanism downstream

of CDH4 as other cadherins, such as CDH1, are known to regulate Wnt signaling[33,34]. Indeed, restoration of CDH4 altered the nuclear localization of β-catenin to the cytoplasm and down-regulated known β-catenin target genes including *AXIN2*, *CCND1*, *C-MYC*, *COX2*, and *MMP7* (Fig. 7d–f and Supplementary Fig. 9k), thereby maintaining the epithelial state of the cell. Cytoplasmic fraction of β-catenin is phosphorylated by glycogen synthase kinase 3β (GSK3β), which marks the protein for β-transducin repeat-containing protein (β-TrCP) mediated proteasomal degradation in the absence of an activating ligand for the signaling pathway. Thus, we examined the association of

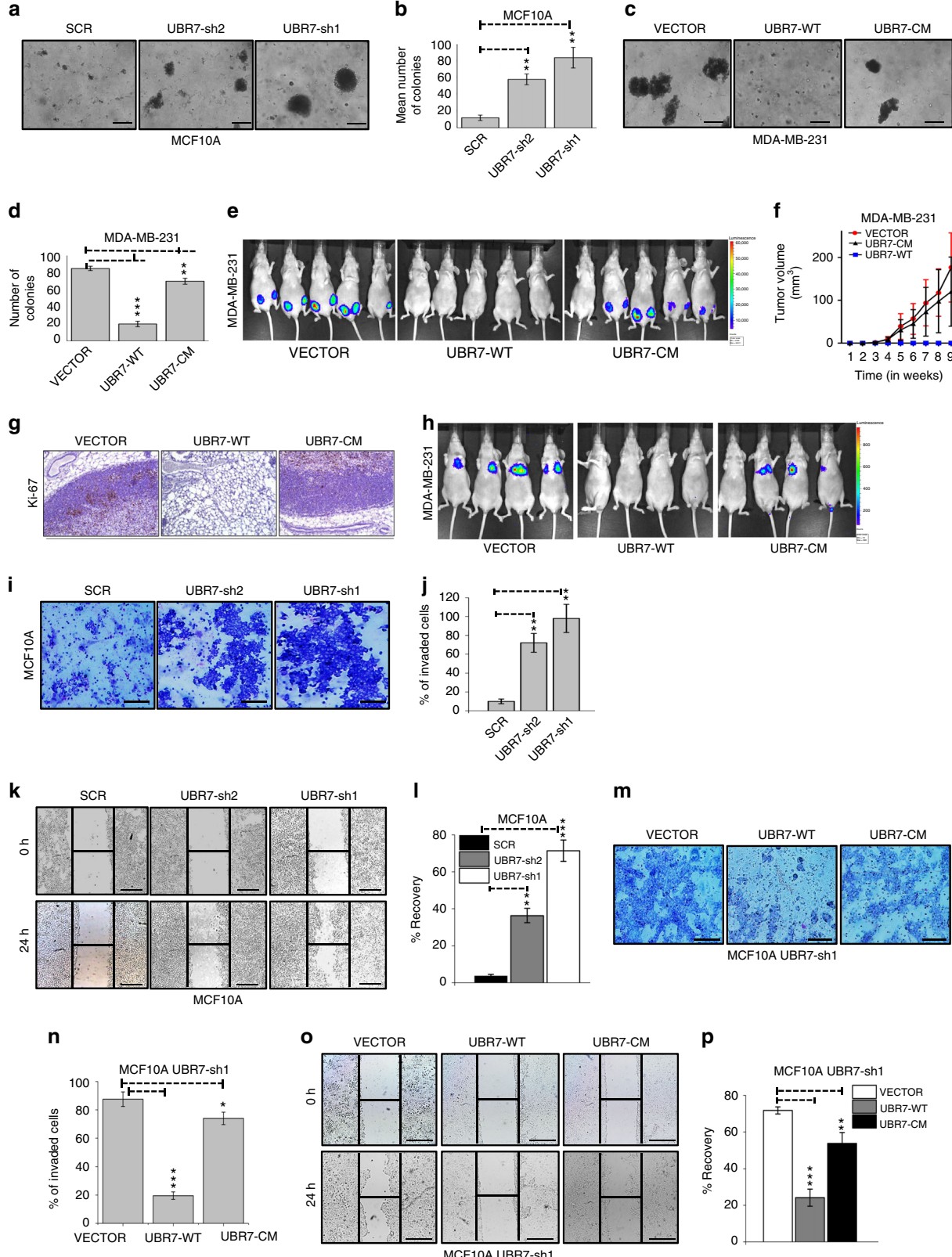

β-catenin with this degradation complex in the absence of UBR7. We found a weak association of this degradation complex, which includes GSK3β and β-TrCP with β-catenin in the absence of UBR7, indicating escape of β-catenin from the degradation pathway, and entry into the nucleus, thereby activating its target genes (Fig. 7g). Overall, our results demonstrated that UBR7 loss activates the Wnt/β-catenin signaling pathway, which is inhibited upon CDH4 restoration.

**Fig. 4** UBR7 acts as tumor metastasis suppressor gene. **a–d** Soft agar assay with MCF10A cells expressing scrambled (SCR) or UBR7-shRNAs (UBR7-sh1 or -sh2) (**a**, **b**) or with MDA-MB-231 cells expressing vector (VECTOR), wild-type (UBR7-WT), or catalytic-mutant (UBR7-CM) (**c**, **d**). **e**, **f** Tumor formation in mice injected into mammary fat pad with MDA-MB-231 cells expressing an empty vector (VECTOR) or WT (UBR7-WT) or CM (UBR7-CM). $n = 5$ mice per group. **g** Immunohistochemistry (IHC) for proliferation marker Ki-67 in tumors derived from control cells versus UBR7-WT- or UBR7-CM-overexpressing cells. Image was taken at ×20 magnification. Scale bar indicates 50 μm. **h** Tumor metastasis in mice injected into tail vein with MDA-MB-231 cells expressing empty vector (VECTOR) or WT (UBR7-WT) or CM (UBR7-CM). $n = 4$ mice per group. **i**, **j** Invaded MCF10A cells expressing scrambled (SCR) or UBR7-shRNAs in a Matrigel chamber were photographed and counted. **k**, **l** Wound healing/migration rate of MCF10A cells expressing scrambled (SCR) or UBR7-shRNAs was monitored. **m**, **n** Invaded MCF10A UBR7-sh1 cells expressing WT (UBR7-WT) or CM (UBR7-CM) in a Matrigel chamber were photographed and counted. **o**, **p** Wound healing/migration rate of MCF10A UBR7-sh1 cells expressing WT (UBR7-WT) or CM (UBR7-CM) was monitored. In **a**, **c**, **i**, **k**, **m**, **o** scale bar indicates 10 μm. In **b**, **d**, **j**, **l**, **n**, and **p**, error bars indicate standard deviation (s.d.); $n = 3$ technical replicates of a representative experiment (out of three experiments). In **f**, error bars indicate standard deviation (s.d.); $n = 5$ mice per group. $P$ values were calculated using two-tailed $t$ tests. *$P < 0.05$; **$P < 0.001$; ***$P < 0.0001$

## Discussion

PHD fingers are structurally conserved chromatin-binding modules present in proteins that are associated with chromatin and regulate gene transcription[35]. The versatile function of a PHD finger as an epigenome reader promotes the recruitment of multi-protein complexes to change the chromatin structure, thereby augmenting transcription (initiation, elongation, and termination) or its repression[35]. A PHD finger fold consists of a C-terminal α-helix and two strands of anti-parallel β-sheet, with a conserved Cys4-His-Cys3 motif in a cross-brace topology, which anchors two zinc atoms[36]. PHD fingers are well known for reading methylated H3 lysine 4 or unmethylated N-term tail of H3, with a few exceptions. The RING finger, which is well characterized for its E3 ubiquitin ligase activity, and has a Cys3-His-Cys4 motif organization, is reciprocal to the PHD finger[37]. The UBR7-PHD finger, on the other hand, has a unique motif organization (Cys4-His2-Cys2) different from that of a canonical PHD or RING finger and hence can be considered as an "atypical" PHD finger. We report here an E3 ubiquitin ligase activity by a PHD finger of UBR7 toward histone H2B at lysine 120. Of note, in comparison with other E3 ubiquitin ligases, which act in a complex[25], UBR7 can promote enzymatic catalysis in isolation. Indeed, mutating the two His residues, which is instrumental to anchoring the zinc coordination complex, abrogates the catalytic activity. Structural studies will provide more insight into the molecular mechanism of H2BK120Ub catalysis by this atypical domain. Also, a detailed investigation of its reader function could provide more insight into the role of the UBR7-PHD finger in cellular context.

UBR7 loss not only drastically reduced H2BK120Ub but also significantly reduced H3K79Me2 without much effect on H3K4Me3, H3K27Ac, H3K27Me3, and H3K9Me3. This was intriguing because several studies have suggested that H2BK120Ub marked nucleosomes act as templates for DOT1L and SET/COMPASS complexes for making H3K79Me2 and H3K4Me3, respectively[38–40]. Interestingly, in the chromatin state analyses, cells with UBR7 loss also harbored changes in transcription states that had predominance of H3K79Me2. Since H3K79Me2 mark is linked with transcriptional elongation and observed on genes that are being actively transcribed, we propose that UBR7 may play important roles in transcriptional elongation. Further biochemical studies focusing on isolation of UBR7 protein complexes or identification of interacting partners will be needed to determine its exact function in this process, if any.

Importantly, we found UBR7 loss to be highly correlated with triple-negative and basal-like breast cancer. Although the molecular features that define triple-negative breast cancer (loss of ER, PR, and Her2 expression) are clear, ambiguity exists in the definition of basal-like cancers. UBR7 loss may be a key determining

feature of this aggressive subtype of breast cancer. Notably, we established that UBR7 can suppress breast tumor formation and metastasis in vivo. One possibility of such an event is through UBR7's prevention of a "self-seeding" event, in which local invading cancer cells lead to tumor formation via fusion of propagating colonies[41]. Also, it is difficult to determine the relative contribution of proliferative versus invasive role for UBR7 in its pro-metastatic function. Overall, UBR7 loss may be a predictive biomarker and provide specific vulnerabilities to epigenetic inhibitors given its drastic impact on chromatin states. Furthermore, given the substantial impact of UBR7 on maintenance of the epithelial state and inhibition of the plasticity of a cell, determining its role in suppression of other malignancies requires further investigation.

We demonstrated that CDH4/R-cadherin is a major target downstream of UBR7. Cadherins are crucial in the maintenance of cell boundary, tissue morphogenesis, and cell polarity[42], and aberrant function may lead to severe metastatic neoplasia[33]. Whereas UBR7 loss promotes breast tumor metastasis, CDH4 overexpression provides only a partial rescue of such phenotypes. CDH4 was found to be highly expressed in mammary epithelial cells, but severely down-regulated in invasive ductal carcinoma[31]. Moreover, due to heterogeneity of cancer cells, CDH4 was absent in those cells that were poorly differentiated[31]. Although we showed that UBR7 prevents metastatic colonization and represses genes that are crucial for breast cancer metastasis by targeting CDH4, other downstream pathways and target genes may also have important roles. For example, other proteins, such as integrins and cytokines, have already demonstrated roles in breast cancer etiology and metastasis[43].

Our results indicate that CDH4 may control β-catenin signaling in a manner similar to E-cadherin-mediated control of β-catenin signaling, which is well documented to play important roles in metastasis. Loss of E-cadherin leads to β-catenin release from the cell surface, resulting in escape from the β-TrCP-mediated degradation pathway, and thereby promoting its nuclear localization and activation of target genes[32,34,44]. In UBR7-depleted cells, Wnt/ β-catenin signaling was one of the top misregulated pathway, and overexpression of CDH4 downregulates Wnt signaling in UBR7-depleted cells by reinstating β-catenin to the cell membrane and reducing the nuclear pool. In addition, we observed that several other Wnt signaling regulators (such as WNT3A, FZD2/3, LRP5/6, ROR2, DSH2, and DDK1) were transcriptionally controlled by UBR7, strongly suggesting that UBR7 is a key mediator of the Wnt/β-catenin signaling cascade. Collectively, our results demonstrated that UBR7 is a H2B E3 ubiquitin ligase that suppresses triple-negative subtype of breast cancer by activating CDH4/R-cadherin expression and inhibiting the canonical Wnt/β-catenin signaling pathway.

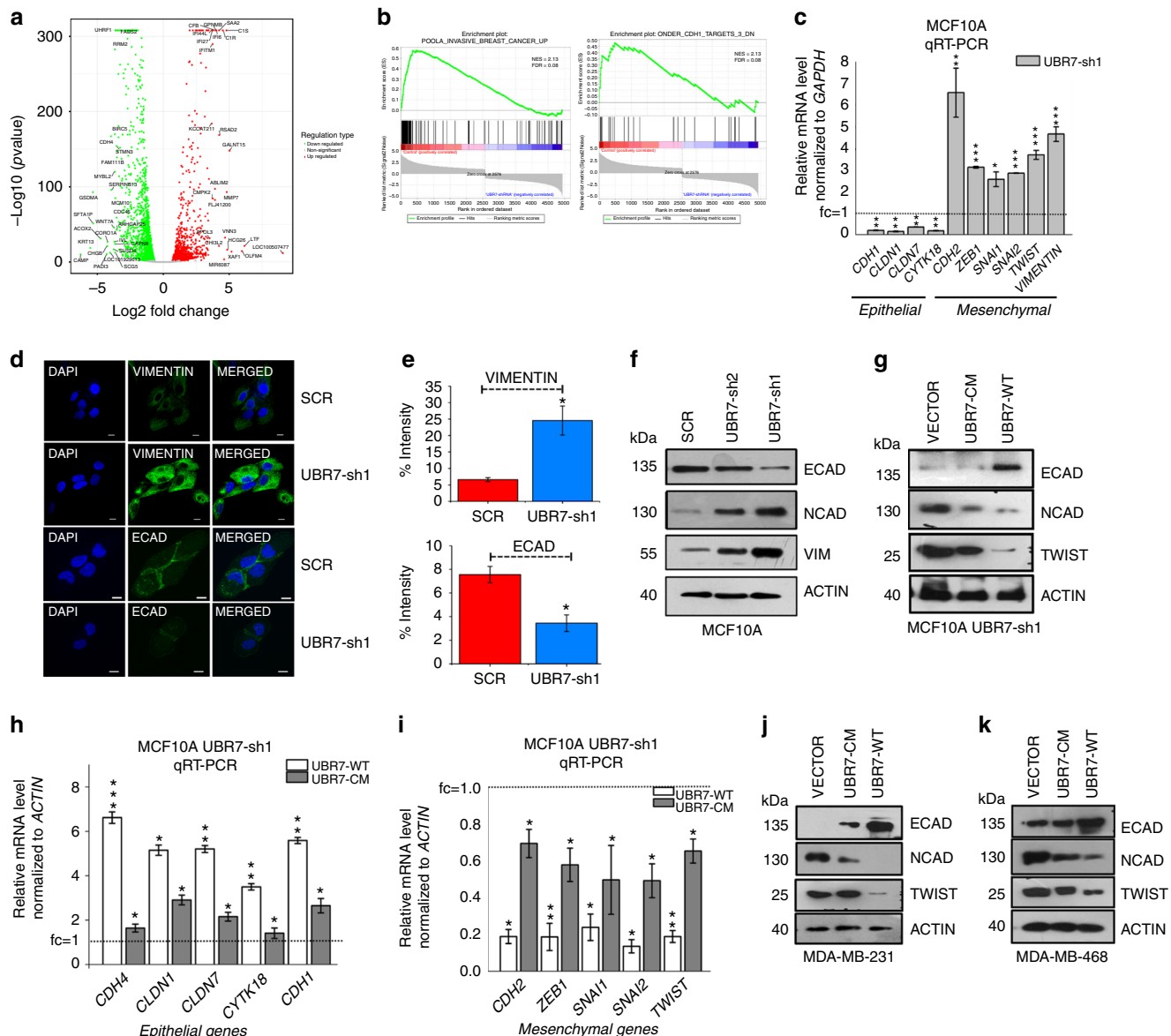

**Fig. 5** UBR7 is an epithelial-to-mesenchymal transition (EMT) suppressor. **a** Volcano plots of the differentially expressed genes upon knockdown of UBR7 (using UBR7-sh1) in MCF10A cells scored via RNA-sequencing (RNA-seq) analysis. Green, red and grey indicates down-, up-regulated and non-significant, respectively. **b** Gene Set Enrichment Analysis (GSEA) output images of two chosen pathways displaying a correlation of differentially regulated genes in UBR7-knockdown MCF10A cells with the "poola_invasive_breast_cancer_upreguated" set and "Onder_CDH1_targets_downregulated" set. **c** Quantitative real-time PCR (qRT-PCR) analysis of EMT signature gene expression upon UBR7 knockdown (UBR7-sh1) in MCF10A cells. **d, e** Immunofluorescence studies showing expression of VIMENTIN and ECAD after UBR7 knockdown (UBR7-sh1) in MCF10A cells (**d**). Scale bar indicates 100 μm. Percent intensity is quantified (**e**). **f, g** Immunoblots of MCF10A cells expressing scrambled (SCR) or UBR7 short hairpin RNAs (shRNAs) (UBR7-sh1 and sh2) (**f**) or MCF10A UBR7-sh1 expressing wild-type (UBR7-WT) and catalytic-mutant (UBR7-CM) (**g**), monitoring expression of candidates regulating EMT. ACTIN was used as a loading control. **h, i** qRT-PCR analysis of epithelial (**h**) and mesenchymal (**i**) genes in MCF10A UBR7-sh1 expressing WT (UBR7-WT) and CM (UBR7-CM). **j, k** Immunoblots of MDA-MB-231 (**j**) and MDA-MB-468 (**k**) cells expressing WT (UBR7-WT) and CM (UBR7-CM) monitoring expression of candidates regulating EMT. ACTIN was used as a loading control. In **c, h, i**, error bars indicate standard deviation (s.d.); $n = 3$ technical replicates of a representative experiment (out of three experiments). P values were calculated using two-tailed $t$ tests. *$P < 0.05$; **$P < 0.001$; ***$P < 0.0001$

## Methods

**Cell lines and cell culture**. HEK293T cells were maintained in Dulbecco's modified Eagle's medium (DMEM; Gibco) supplemented with 10% fetal bovine serum (FBS; Gibco) and 1% antibiotic–antimycotic (Gibco) at 37 °C and 5% $CO_2$. MCF10A and MCF12A cells were maintained in DMEM/Ham's F12 supplemented with 5% horse serum (Gibco), epidermal growth factor (EGF), insulin, hydrocortisone, cholera toxin (Sigma), and 1% antibiotic–antimycotic. MDA-MB-231, MDA-MB-468, MCF7, and T47D cells were maintained in RPMI-1640 medium (Gibco) supplemented with 10% FBS, insulin, and 1% antibiotic–antimycotic. All cell lines were purchased from ATCC. 16N, 21PT, and 21MT2 (provided by R. Weinberg, Whitehead Institute, Massachusetts Institute of Technology) were maintained in DMEM supplemented with 10% FBS, insulin, hydrocortisone, EGF,

and 1% antibiotic–antimycotic. 4T1 and 4T07 (provided by R. Weinberg) were maintained in DMEM/Ham's F12 medium supplemented with 10% FBS, insulin, hydrocortisone, and 1% antibiotic–antimycotic. All cell lines used in the study were negative for mycoplasma. All cell lines were validated by MD Anderson Cancer Center Characterized Cell line core facility via DNA fingerprinting. For transient transfection, cells were counted and seeded in 12-well or 6-well or 6-cm dishes and then subjected to overexpression using Lipofectamine-2000 (Invitrogen) as per the manufacturer's protocol.

**Protein purification**. The full-length *UBR7* complementary DNA (cDNA) sequence, UBR domain alone, or PHD finger alone was cloned in a pDEST15

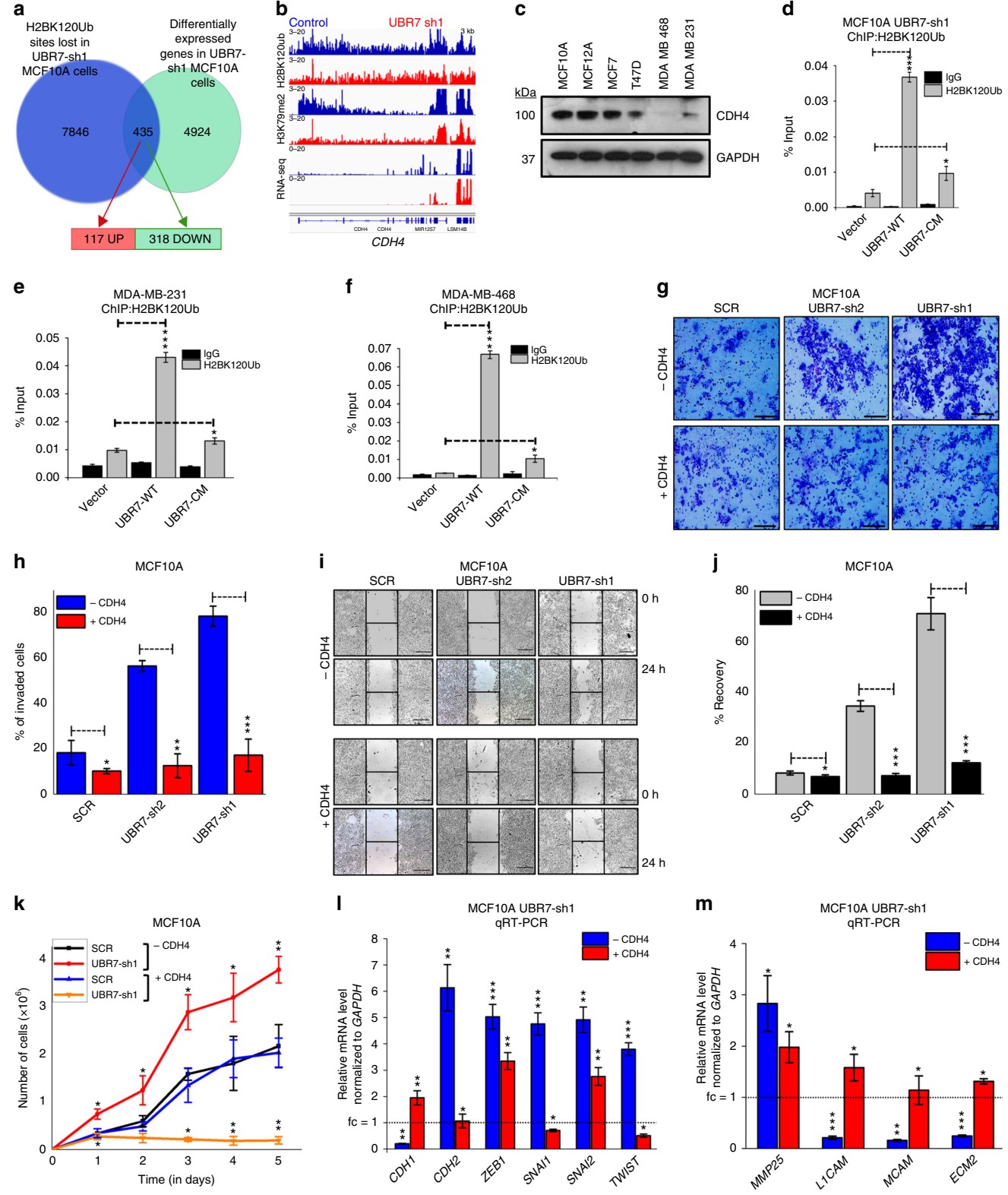

vector (GATEWAY cloning system, Invitrogen) and sequence verified. The protein used in in vitro assays were purified as described previously[45]. Briefly transformed cells were grown till 0.8 optical density (OD) and induced with 1 mM isopropyl β-D-1-thiogalactopyranoside (Sigma) at 20 °C for 16 h. The pellets were resuspended in lysis buffer and lysed mildly followed by glutathione sepharose beads binding and washing with wash buffer. The proteins were eluted and purified further to homogeneity via gel filtration chromatography using a Superdex75 column (GE Healthcare). CM derivative (H163S/H166S) were generated using a QuikChange site-directed mutagenesis kit (Stratagene) as per the standard protocol[46].

**Nucleosome isolation**. Nucleosomes were prepared freshly from HeLa cells as described previously[45]. Briefly, the nuclear pellet from HeLa cells was digested with MNase (0.2 Units/μl; Sigma) and extracted with TE buffer for 1 h. The mononucleosome was separated via sucrose gradient (5–40%) ultracentrifugation using Sorvall WXUltra100 (Thermo Fischer Scientific) with AH650 rotor for 16 h at 207,203 × g. For further analysis, these fractions were pooled and concentrated.

**In vitro ubiquitination assay**. In vitro ubiquitination reactions were set up with purified UBR7-PHD, UBR7-WT, or UBR7-CM (H163S/H166S) as E3 enzymes and

**Fig. 6** UBR7 suppresses epithelial-to-mesenchymal transition (EMT) through activation of CDH4. **a** Venn diagram showing overlaps of H2BK120Ub enriched and differentially regulated genes after UBR7 knockdown. **b** Integrative Genomics Viewer (IGV) view of H2BK120Ub and H3K79me2 chromatin immunoprecipitation sequencing (ChIP-seq) and RNA-sequencing (RNA-seq) tracks on the *CDH4* gene in Control (SCR) or UBR7-sh1 short hairpin RNA (shRNA) expressing MCF10A cells. **c** Immunoblot showing the expression of CDH4 across different normal breast cell and breast cancer cell lines. Glyceraldehyde 3-phosphate dehydrogenase (GAPDH) was used as a loading control. **d–f** Bar plots showing H2BK120Ub ChIP in the *CDH4* gene locus in MCF10A UBR7-sh1 (**d**) or MDA-MB-231 (**e**) or MDA-MB-468 (**f**) cells expressing wild-type (UBR7-WT) and catalytic-mutant (UBR7-CM). **g, h** Invasion of MCF10A cells overexpressing CDH4 in the presence (SCR) and absence (UBR7-sh1 and UBR7-sh2) of UBR7 in a Matrigel chamber was photographed and quantitated. **i, j** Wound healing by MCF10A cells overexpressing CDH4 in the presence (SCR) and absence (UBR7-sh1 and UBR7-sh2) of UBR7 was photographed and the percent recovery was measured over time. **k** Proliferation of cultured MCF10A cells overexpressing CDH4 in the presence (SCR) and absence (UBR7-sh1) of *UBR7*. **l, m** Quantitative real-time PCR (qRT-PCR) analysis of EMT signature genes (**l**) and cell adhesion-linked genes (**m**) in UBR7-sh1 MCF10A cells upon CDH4 overexpression. In **g, i**, scale bar indicates 10 µm. In **d–f, h, j–m**, error bars indicate standard deviation (s.d.); $n = 3$ technical replicates of a representative experiment (out of three experiments). *P* values were calculated using two-tailed *t* tests. *$P < 0.05$; **$P < 0.001$; ***$P < 0.0001$

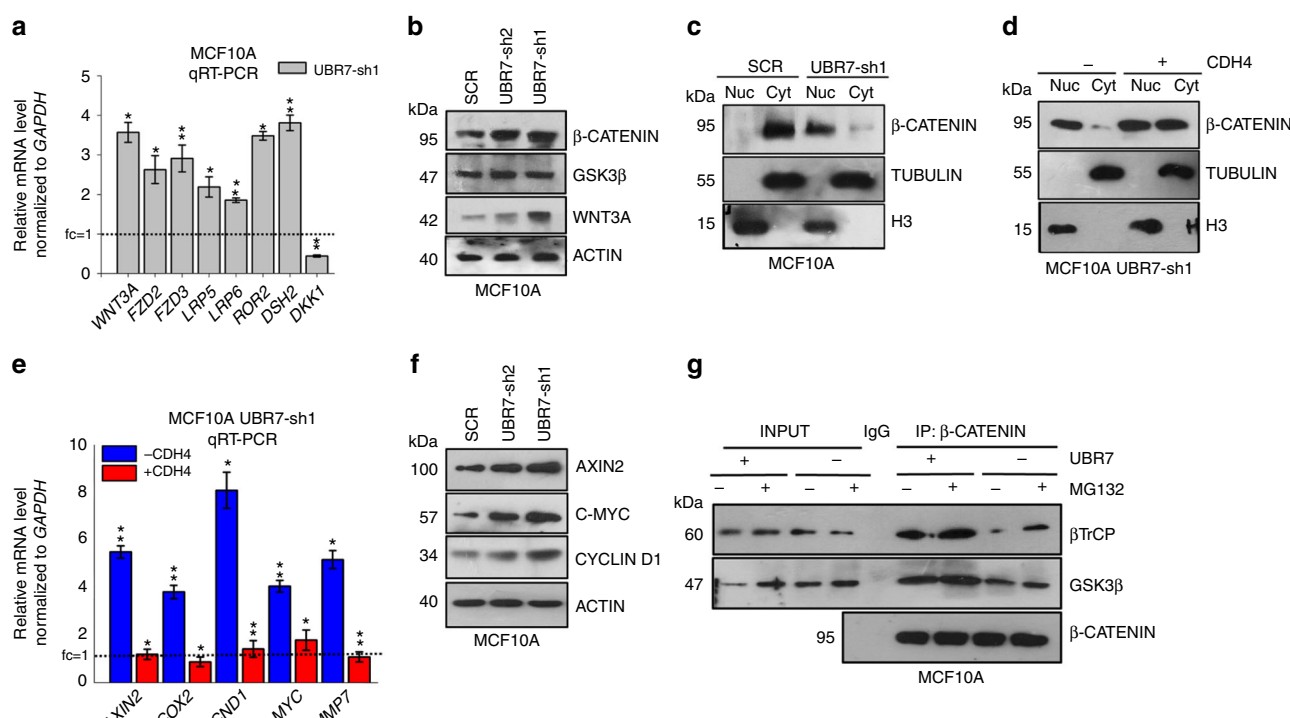

**Fig. 7** UBR7 silencing activates the Wnt/β-catenin signaling pathway. **a** Quantitative real-time PCR (qRT-PCR) analysis of key regulators of the Wnt/β-catenin signaling pathway upon loss of UBR7 (UBR7-sh1). **b** Immunoblots monitoring expression of β-CATENIN, glycogen synthase kinase 3β (GSK3β), WNT3A, and ACTIN (loading control) in MCF10A cells expressing scrambled (SCR) or UBR7-shRNAs. **c** Immunoblots monitoring expression of β-CATENIN in the nuclear and cytoplasmic fractions of MCF10A cells expressing scrambled (SCR) or UBR7-sh1shRNA. Histone H3 and TUBULIN were used as loading controls for nuclear and cytoplasmic lysates, respectively. **d** Immunoblots monitoring expression of β-CATENIN in the nuclear and cytoplasmic fractions of *CDH4*-expressing MCF10A UBR7-sh1 cells. Histone H3 and TUBULIN were used as loading controls for nuclear and cytoplasmic lysates, respectively. **e** qRT-PCR analysis of β-catenin target genes expression in UBR7-sh1 MCF10A cells upon CDH4 overexpression. **f** Immunoblots showing expression of β-catenin target genes upon UBR7 knockdown (UBR7-sh1 or -sh2) in MCF10A cells. **g** Co-immunoprecipitation of β-catenin from UBR7-knockdown MCF10A cells in the presence or absence of MG132 (20 µM for 18 h) showing association of β-catenin with GSK3β and β-transducin repeat-containing protein (β-TrCP). In **a, e**, error bars indicate standard deviation (s.d.); $n = 3$ technical replicates of a representative experiment (out of three experiments). *P* values were calculated using two-tailed *t* tests. *$P < 0.05$; **$P < 0.001$

recombinant H2B, H2A/H2B dimer, core histones octamer, or purified nucleosome as a substrate using Ubiquitinylation kit (Cat #: BML-UW9920, Enzo Life Sciences) as per the manufacturer's protocol. Briefly, the reaction was carried out in ubiquitinylation buffer containing 100 U/ml inorganic pyrophosphatase, 1 mM dithiothreitol (DTT), and 5 mM EDTA (negative control). 2.5 µM of ubiquitin (biotinylated) was incubated with 100 nM E1, 2.5 µM E2 (for UBR7, UbcH6 acts as the E2), 5 mM Mg-ATP, and 100 nM E3 along with 1 µM substrate at 37 °C for 1 h. The reaction was stopped, and trichloroacetic acid precipitation was performed and analyzed via western blotting with antibodies against H2BK120Ub, H2B, H2A, H3, and H4 antibodies.

**Peptide pull-down assay.** A peptide pull-down assay was performed as described previously[45]. Briefly, equivalent amounts of peptides and protein were incubated in immunoprecipitation (IP) buffer (50 mM Tris, pH 7.5, 150 mM NaCl, 0.05%

NP-40, 1 mM DTT). The complex was pulled down with streptavidin beads, washed with the IP buffer, and eluted and analyzed using western blotting.

**GST pull-down assay.** GST and GST-fusion proteins were incubated with the recombinant histones H3, H4, H2A, and H2B at equimolar ratios in a IP buffer overnight as described previously[45]. The complex was pulled down with glutathione sepharose bead (GE Healthcare), washed with IP buffer, and analyzed using western blotting with specific antibodies. Ten percent of the histone proteins (H3, H4, H2A, and H2B) were used as inputs.

**Co-immunoprecipitation.** Cells were subjected to co-immunoprecipitation as delineated previously[45]. In brief, cells were lysed in lysis buffer (20 mM Tris pH 8.0,

150 mM NaCl, 1% NP-40, 0.5% sodium deoxycholate, 0.1% sodium dodecyl sulfate (SDS), 1 mM EDTA) and pulled down with specific antibodies or FLAG M2 beads, followed by washes with the lysis buffer. The immunoprecipitant was analyzed using western blotting. Ten percent of the lysate with which IP was set was used as the input.

**RNA interference through lentiviral production**. shRNA plasmids for UBR7 with pLKO.1-puro backbone (Sigma-Aldrich) were screened for efficient knockdown. Two of seven shRNAs were selected for subsequent experiments. Their sequences are as follows: UBR7-sh1 5′-CAGTGCACCCAGGGGTTATTTG-3′ and UBR7-sh2–5′-GCTTAAAGCTAAGCAGCTTAT-3′. UBR7-sh1 targets the 3′-UTR of the gene, so the overexpression constructs were resistant to the shRNAs. HEK293T cells were plated at a density of $3 \times 10^5$ cells in 10-cm dishes. Eight micrograms of shRNA and packaging vectors were transfected into the cells as described previously[47]. Transduced cells were selected using puromycin (10 μg/ml; Sigma) for 3 days.

**UBR7 overexpression via lentiviral production**. WT and H163S/H166S CM of UBR7 were cloned into the pHAGE-CMV-fullEF1a-IRES-ZsGreen (from Jeng-Shin Lee; Dana-Farber/Harvard Cancer Center). 293T cells were plated at density of $3 \times 10^5$ in 10-cm dishes. Recombinant lentiviral particles were produced using Lipofectamine 2000-mediated transient transfection in HEK293T cells. Briefly, 8 μg of overexpression vectors, a packaging vector (psPAX2), and an envelope vector (pMD2.G) were transfected into 293T cells plated in 10-cm dishes. The viral supernatant was harvested 48 and 72 h after transfection and filtered. Cells were infected three times in 48 h with the viral supernatant containing 10 μg/ml Polybrene. Green fluorescent protein-positive cells were sorted and cultured for other experiments.

**Quantitative real-time PCR**. Total RNA was isolated using TRIzol reagent (Invitrogen) and reverse transcribed using a Revertaid First Strand cDNA Synthesis kit (Thermo Fischer Scientific) according to the manufacturer's protocol followed by qRT-PCR using ABI-SYBR GREEN mix (Applied Biosystems). qRT-PCR was performed using StepONE plus FAST Real-time PCR machine. Each sample was analyzed independently three times and the results of one representative experiment, with technical triplicates, are shown. List of primers is provided in Supplementary Table 1.

**Western blot analysis**. Whole-cell lysates were prepared in lysis buffer (20 mM Tris, pH 8.0, 150 mM NaCl, 1% NP-40, 0.5% sodium deoxycholate, 0.1% SDS, 1 mM EDTA) as described previously[45]. The lysates were electrophoresed on 7.5%, 11%, or 15% SDS-polyacrylamide gel electrophoresis gels. Blots were probed with specific antibodies. The membrane was blocked with 5% bovine serum albumin (BSA) or non-fat dry milk in TBST (Tris-buffered saline, 0.1% Tween-20) . The antibodies used are listed in Supplementary Table 2. All uncropped images of the blots are provided in Supplementary Fig. 10.

**Immunofluorescence**. Immunofluorescence was carried out as described previously[45]. Cells were fixed with 4% paraformaldehyde or methanol, permeabilized with 1% Triton X-100 or 100% methanol, blocked with 3% BSA, and stained with the indicated antibodies. Coverslips were mounted after staining with DAPI (4′,6-diamidino-2-phenylindole) and photographed using a Nikon T1E confocal microscope with an A1RMP Scanner Head.

**Migration or wound healing assay**. Migration assay was performed as described previously[48]. Cells were plated in 6-well dish in triplicates and grew to 90% confluence and the cell surface was scratched with a sterile 20 μl tip. Images were captured with Nikon T1 E100 microscope at 0 h and post 24 h after the scratch to measure the wound healing by the cells, indicating their migratory capacity and migration rate. The images were analyzed using the Image J software program (National Institutes of Health; NIH), to measure the wound recovery by the cells indicating their migratory ability.

**Proliferation assay**. Transduced and transfected cell lines were plated at a density of $1 \times 10^6$ cells in triplicates in a 24-well plate. MTT (3-[4,5-dimethylthiazol-2-yl]-2,5 diphenyl tetrazolium bromide) was added at indicated time points and MTT assay was performed as described previously[49]. The OD was measured using an Epson Plate reader. The number of proliferating cells was calculated from the standard curve.

**Invasion assay**. Invasion assay was performed as described previously[47]. For this assay, $1 \times 10^6$ cells were cultured in the upper well of Matrigel chambers (Corning) containing serum-free medium and allowed to migrate towards serum-enriched medium in the bottom wells. After 20 h of incubation, invading or migrating cells were stained with 0.5% Giemsa, photographed with Nikon T1 E100 microscope, and counted using the Image J software program.

**Soft agar assay**. Transduced human breast epithelial and cancer cell lines were resuspended in 3 ml of soft agar (medium containing 0.3% noble agar [Affymetrix]) warmed to 45 ℃. The cell suspension was layered onto 3 ml of bottom agar (medium containing 0.8% noble agar) in a 6-well plate (six replicates). 2 ml of the medium was added to the top agar and changed every 3 days. Visible colonies were scored after 4–5 weeks and stained with 0.5% Giemsa. The number of colonies and mean area of colonies was calculated using the Image J software program.

**Animal experiments and in vivo imaging**. Animals used in these experiments were all female nude mice aged 8–10 weeks (The Jackson Laboratory). All cells used for in vivo injections were labeled with red fluorescent protein luciferase. The mice were anesthetized with isoflurane and injected with $2 \times 10^6$ MDA-MB-231 cells in 50 μl of phosphate-buffered saline (PBS) in the abdominal mammary fat pads on both sides ($n = 5$ mice). Tumor growth was monitored weekly via caliper measurement and bioluminescent imaging once every 2 weeks. Once the largest tumor diameter was reached (1.5 cm, which is the maximal tumor diameter allowed under our institutional protocol), the animals were sacrificed. For the tail vein injections, the mice ($n = 4$) were injected with 100 μl of $1 \times 10^6$ cells were injected into the medial tail vein. All mouse experiments were performed with the approval of the MD Anderson Institutional Animal Care and Use Committee.

For in vivo imaging, cells were infected with EF1-RFP-T2A-Luciferase (System Biosciences) to enable stable expression of firefly luciferase. All in vivo bioluminescent imaging was carried out at the MD Anderson Small Animal Imaging Facility. For this imaging, the animals were anesthetized with isoflurane. They were injected intraperitoneally with 3 mg of D-luciferin (Perkin Elmer) and imaged using the IVIS Spectrum Imaging System (Perkin Elmer). Analysis after acquisition was done using the Living Image software program (version 4.3; Perkin Elmer).

**Tissue microarrays**. Patients: Breast tumors and normal breast tissue were obtained from patients who underwent surgery at MD Anderson Cancer Center from 2005 to 2015. Information on their hormone and Her2 statuses, as well as proliferation fraction (Ki-67), was retrieved from pathological and clinical reports. ER information was available for 367 (99%) patients, PR information was available for 368 (99%) patients, Her2 information was available for 362 (98%) patients, and proliferation fraction (Ki-67) information was available for 208 (56%) patients.

Breast cancer TMAs from MD Anderson Cancer Center and a commercially available TMA BR2082a (US Biomax, Inc. Rockville, MD, USA) were used. The TMAs from MDACC included 538 breast cancers and 15 normal tissues. Breast cancers were represented in 389 (60%) cases with 3 punches of 1 mm, in 96 (17.8%) cases with 3 punches of 0.6 mm, and in 53 cases (9.9%) with 3 punches of 1 mm and 3 punches of 0.6 mm. The normal tissue samples included six punches of 0.6 mm from three normal lymph nodes, three normal breast tissues, three normal kidneys, three normal colon, and three normal lung samples. The composition of the commercial TMA is shown at the vendor's website (http://www.biomax.us/tissue-arrays/Breast/BR2082a).

**Immunohistochemistry**. For validation of the anti-UBR7 antibody, Western blot of UBR7 expression on the whole cell line lysate of MCF10A transfected with a control small interfering RNA (siRNA) and siRNA against UBR7, and of MCF7 transfected with siRNA against UBR7 was performed. These cell lines were paraffin embedded and the final protocol was established to perform staining on 5-μm-thick TMA sections. In brief, epitope retrieval was performed using citrate buffer at pH 6.0 for 20 min, followed by peroxidase blocking for 5 min. A polyclonal rabbit anti-UBR7 antibody (Bethyl Laboratories) was then incubated for 60 min using a 1:2000 dilution followed by polymer (goat-anti-rabbit immunoglobulin G (IgG)) and 3,3′-diaminobenzidine incubation for 8 and 10 min, respectively. Slides were counterstained with hematoxylin. The staining was performed using supplies and an autostainer from Leica Biosystems.

For evaluation of UBR7 staining, nuclear staining was scored semi-quantitative providing the percentage of stained cells and staining intensity (0 = no staining, 1 + = weak staining, 2 + = moderate staining, and 3 + = strong staining). Representative staining examples are shown in Supplementary Fig. 4a.The study was approved by the MD Anderson Institutional Review Board.

**ChIP assay**. ChIP assays were performed as described earlier[50]. Cells were cross-linked with 1% formaldehyde and the chromatin was sheared and immunoprecipitated with the UBR7 antibody (Bethyl Laboratories), H2BK120Ub antibody (Millipore), H2B antibody (Abcam), or as a negative control IgG. ChIP DNA was analyzed by qPCR using gene specific primers. Each ChIP experiments were performed three independent times with technical triplicates.

**ChIP-seq assay**. ChIP assays were performed as described previously[50] with minor modifications. Briefly, ~$2 \times 10^7$ cells were harvested via cross-linking with 1% (wt/vol) formaldehyde for 10 min at 37 ℃ with shaking. After quenching with 150 mM glycine for 10 min at 37 ℃ with shaking, cells were washed twice with ice-cold PBS and frozen at −80 ℃ for further processing. Cross-linked pellets were thawed and lysed on ice for 30 min in ChIP harvest buffer (12 mM Tris-Cl, 1 × PBS, 6 mM EDTA, 0.5% SDS) with protease inhibitors (Sigma). Lysed cells were sonicated with

a Bioruptor (Diagenode) to obtain chromatin fragments (~200–500 bp) and centrifuged at $15,000 \times g$ for 15 min to obtain a soluble chromatin fraction. In parallel with cellular lysis and sonication, antibodies ($5 \mu g/3 \times 10^6$ cells) were coupled with 30 μl of magnetic protein G beads in binding/blocking buffer (PBS + 0.1% Tween + 0.2% BSA) for 2 h at 4 °C with rotation. Soluble chromatin was diluted five times using ChIP dilution buffer (10 mM Tris-Cl, 140 mM NaCl, 0.1% dissolved organic compound, 1% Triton X, 1 mM EDTA) with protease inhibitors and added to the antibody-coupled beads with rotation at 4 °C overnight. After washing, samples were treated with elution buffer (10 mM Tris-Cl, pH 8.0, 5 mM EDTA, 300 mM NaCl, 0.5% SDS), RNase A, and Proteinase K, and cross-links were reversed overnight. ChIP DNA was purified using AMPure XP beads (Agencourt) and quantified using the Qubit 2000 (Invitrogen) and Bioanalyzer 1000 (Agilent). Libraries for Illumina sequencing were generated following the New England BioLabs (NEB) Next Ultra DNA Library Prep Kit protocol. A total of 10 cycles were used during PCR amplification for the generation of all ChIP-seq libraries. Amplified ChIP DNA was purified using double-sided AMPure XP to retain fragments (~200–500 bp) and quantified using the Qubit 2000 and Bioanalyzer 1000 before multiplexing.

**ChIP-seq data processing**. Raw fastq reads for all ChIP-seq experiments were processed using FastQC (http://www.bioinformatics.babraham.ac.uk/projects/fastqc/), and quality reads were aligned to the hg19 reference genome using Bowtie version 1.1.2[51] with the following criteria: -n 1 -m 1–best–strata. Duplicate reads were marked using SAMBLASTER[52] before compression to BAM files. To directly compare Control and UBR7-shRNA ChIP-seq samples, uniquely mapped reads for each mark were normalized by total reads per condition, sorted, and indexed using samtools version 0.1.19[53].

Model-based analysis of ChIP-seq (MACS) (version 1.4.2; peak calling algorithm with a $p$ value threshold of $1e-7$)[54] was used to identify H2BK120Ub enrichment over "input" background. Unique H2BK120Ub binding sites were identified using the concatenate, cluster, and subtract tools from the Galaxy/Cistrome web-based platform[55]. Briefly, a shared peak set was first generated by clustering intervals of H2BK120-Control peaks that directly overlapped H2BK120-UBR7-shRNA peaks by a minimum of 1 bp. Unique peaks were then identified by subtracting the total number of H2BK120 peaks in each condition by the shared peak set. Venn diagrams were generated using the Venn Diagram tool in Galaxy. To visualize ChIP-seq libraries on the IGV browser, we used deepTools version 2.4.060 to generate bigWig files by scaling the bam files to reads per kilobase per million (RPKM) using the following criteria: bamCoverage –b–normalizeUsing RPKM–smoothLength 300–binSize 30–extendReads 200 –o.

A list of known genes was obtained from the UCSC Genome browser (http://genome.ucsc.edu/). Proximal promoters were defined as ±5 kb from the transcription start site (TSS) and the genebody was defined as all genic regions outside of the +5 kb promoter region. Intergenic regions were defined as all regions outside both the proximal promoter and genebody. H2BK120 peaks were assigned to genes if they overlapped the promoter or genebody by a minimum of 1 bp. These H2BK120Ub "enriched" regions were further used for the generation of read density plots for all ChIP-seq data. All read density plots were generated using thengs.plotpackage in R[56].

**ChIP-seq analysis for UBR7, RNF20, and RNF40 comparison**. To directly compare Control SCR, UBR7, RNF20, and RNF40 shRNA ChIP-seq samples, uniquely mapped reads for H2BK120Ub in all conditions were normalized to ~10 million reads. For generation of Supplementary Fig. 3b–d, Control replicate1 was used for the UBR7-shRNA-1 comparison and Control replicate2 was used for the RNF20/RNF40 shRNA comparisons. Normalization of Control replicate1 and UBR7-shRNA-1 samples to ~10 million reads displayed little effect on the average density profile of H2BK120Ub (Supplementary Fig. 3b). For further UBR7, RNF20, and RNF40 comparisons, read counts for Control (SCR) H2BK120ub and Input replicates were merged together, normalized to ~10 million reads, and peaks were called using MACS ($p$ value $1e-7$). To identify unique H2BK120 binding sites that were lost upon knockdown of either UBR7, RNF20, or RNF40, a shared peak set was first generated by clustering intervals of Control (SCR) H2BK120Ub sites that directly overlapped either UBR7, RNF20, or RNF40 shRNA H2BK120Ub peaks by a minimum of 1 bp. Unique peaks were then identified by subtracting the total number of H2BK120Ub peaks in each condition by their associated shared peak set. A final Control (SCR) shared peak set was further generated using the Control (SCR) H2BK120Ub binding sites from Supplementary Fig. 3e–g, which were lost from depletion of each factor (UBR7, RNF20, or RNF40). A final Control (SCR) unique peak set was identified by subtracting the total number of H2BK120Ub peaks in each condition by the shared peak set. H2BK120Ub peaks were assigned to genes if they overlapped the promoter (±5kbTSS) or genebody by a minimum of 1 bp and these H2BK120Ub "enriched" regions were used for the generation of average density profiles (Supplementary Fig. 3b–d, 3h) and pathway analysis (Supplementary Fig. 3i).

**Chromatin state calls**. ChromHMM[57] was used to identify combinatorial chromatin state patterns based on the histone modifications studied. Normalized bam files were converted into binarized data at a 1000 bp resolution using the

BinarizeBam command with a $p$ value cut-off of $1e-5$. We specified that ChromHMM should learn a model based on 10 chromatin states. As we considered models between 8 and 20 chromatin states, we chose a 10-state model because it is large enough to identify important functional elements while still being small enough to interpret easily. Overlap enrichment was used to compute differential enrichment in each of the 10 chromatin states between Control and UBR7-shRNA samples. The ChromHMM segment files from the 10-state model contain the genomic locations of each chromatin state called in both the Control and UBR7-shRNA samples. To determine which chromatin states were enriched between conditions, we further compared the genomic locations by using the Control segments file as input for the segment directory, and by further separating the UBR7-shRNA segments file into 10 individual states and using it as input for the external coordinate directory. The UBR7-shRNA segment file was separated into individual chromatin states for the external coordinate directory with the following command:

awk -F/t '{print ≫ $4;close($4)}' ~/path_to/UBR7-sh1_segments.bed

Overlap enrichment was ran using the following command:

java -mx4000M -jar ChromHMM.jar OverlapEnrichment ~/path_to/Control_segments.bed ~/path_to/UBR7-shRNA_segments_separated OverlapEnrichment_Control_vs_UBR7

**RNA-sequencing**. RNA was isolated using RNeasy kit and libraries prepared using Illumina mRNA-Seq library kit. Raw FASTQ reads for all RNA-seq experiments were processed using FastQC and quality reads were aligned with the hg19 reference genome using TopHat(version 2.0.14)[58] with a Bowtie2 (version 2.2.3)[59] index based on UCSC annotations using the following criteria: -G -g 1 -r 150–mate-std-dev 50–library-type fr-unstranded. These criteria preserved only the best reads that uniquely mapped to the genome with one or fewer mismatches. To visualize RNA-seq libraries on the IGV browser, we used deepTools version 2.4.060 to generate bigWig files by scaling the bam files to RPKM using the following criteria: bamCoverage –b --normalizeUsing RPKM --smoothLength 300 --binSize 30 -o.

For identification of differentially expressed genes and gene set enrichment analysis (GSEA), raw counts were obtained by assigning reads at the gene level across the UCSC hg19 reference genome using featureCount[60] in the Rsubread package. DESeq2[61] was employed for normalization and identification of differentially expressed genes in UBR7-shRNA and Control samples. All plots were generated using the ggplot and ggrepel packages in R. GSEA[62] was run with normalized counts from all identified differentially expressed genes using the hallmark, curated, and gene ontology gene sets with default settings.

**Statistical analysis**. TCGA data analysis: Gene expression data for UBR7 from TCGA data were analyzed using the UCSC Xena functional genomics browser. Cancer subtypes for gene expression data were extracted from PAM50 version.

ChIP-qPCR and qRT-PCR: All qRT-PCR, ChIP, and other quantification data were collected in experiments performed in technical triplicate. Each experiment was repeated at least three times, and statistically significant results were obtained. An unpaired two-tailed Student's $t$ test was performed using the Prism software program (GraphPad Software) to assign the significant differences between groups. Significant differences were considered when $P < 0.05$, $*P \leq 0.05$, $**P \leq 0.001$, and $***P \leq 0.0001$. Error bars indicate the standard deviation of the mean for the technical replicates, as indicated in the legend.

TMAs: Two different TMAs were used: one for analyses of primary versus metastatic invasive ductal carcinoma tumors and the other for the association of UBR7 staining with ER, PR, and HER2 status. The intensity and percentage were measured three times for each tumor, and the means of these measures were used for data analyses (missing values were excluded). All statistical analyses were performed using R (version 3.3.1).

**Reporting Summary**. Further information on experimental design is available in the Nature Research Reporting Summary linked to this article.

## Data availability

ChIP-Seq and RNA-Seq data can be accessed at GEO using the accession number: GSE93759. All relevant data are available from the authors upon request.

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

## Acknowledgements

We thank Prof. Robert G. Roeder and Prof. Moshe Oren for providing us with the FLAG H2B wild-type and K120R mutant plasmids and RNF20 shRNA plasmid, respectively. We thank Drs. Michelle C. Barton, Jessica Tyler, and Tapas K. Kundu for critical comments on the manuscript. We acknowledge Scientific Editing team at MD Anderson Cancer Center for proofreading our manuscript. This work was supported in part by research grants from Biomolecular Assembly, Recognition and Dynamics Project (Grant 12-R&D-SIN-5.04-0103) from the Department of Atomic Energy, Swarnajayanti Fellowship, Department of Science and Technology and Ramalingaswami Fellowship, Department of Biotechnology, Government of India to C.D.; CSIR-Network Project (UNSEEN) and Ramanujan fellowship, Department of Science and Technology,

Government of India to S.R.; and National Cancer Institute grant (CA160578) and The University of Texas MD Anderson Cancer Center (start-up funds) to K.R. S.A., A.D., and D.K.S. thank the Council for Scientific and Industrial Research, University Grants Commission and Indian Council for Medical Research, Government of India, respectively, for funding their fellowship. D.C. is supported by the Triumph post-doctoral training program at MD Anderson Cancer Center supported by CPRIT (RP170067).

## Author contributions

S.R., K.R., and C.D. conceived the study, designed experiments, and analyzed the data. S.A., D.C., C.T., I.S., and M.M. designed, performed experiments, and analyzed data. E.T., F.Y., and C.T. stained and scored T.M.A.s. A.A.S. provided reagents. J.M. and R.B. performed statistical analyses related to the TMAs. A.D., D.K.S., C.T., and A.T.R. performed the bioinformatic analysis. C.D., K.R., S.R., S.A., D.C., and C.T. wrote the paper.

## Additional information

**Competing interests:** The authors declare no competing interests.

