## [Peer Review File · Nature Communications]

Reviewers' comments:

Reviewer #1, Expertise: Ubiquitin ligases and cancer (Remarks to the Author):

PHD motifs have been defined to bind to particular modified states of histones, particularly methylated/unmethylated isoforms of specific lysine residues. Adhikary et al have identified another function, ubiquitin ligase activity, for the PHD finger in UBR7. The authors provide both in vitro ubiquitin ligase assays and in vivo data to support this conclusion and to define H2BK120 as a predominant target substrate site. They then go on to define changes in transcription profiles associated with UBR7 knock down in breast cancer cell lines, which leads them to focus on cadherin as a major gene target regulated by UBR7.

The finding that this PHD domain has ub ligase activity is interesting and important. The paper would be strengthened by:

1. A comparison of this PHD domain sequence with other PHD domains, and a prediction of how widespread (or unique) this ub ligase might be. The authors compare the PHD in UBR7 to RING domains found in other UBR proteins, but they do not do such comparisons across PHD motifs. How conserved are the 2 critical histidine residues?
2. Clarification/correction of Fig 2F. As currently drawn, it appears that the shScr control affected H2Bub at 7846 genes, while the UBR7 shRNA affected this modification at ~1100 genes. This appears to be a mistake, as later in the paper the authors overlap the 7800+ genes with changes in expression that occur upon UBR7 knock down. This point is quite important and needs to be clarified.
3. Addition of example histone H2Bub ChIP profiles at specific genes in Fig 2 to show the degree of differences observed more clearly. Currently, the only profile shown is for CDH4 in Fig 6, and the changes in H2Bub do not appear to be that striking.
4. Addition of experiments with the UBR CM construct to the experiments shown in Fig 3 K and I, and also Fig 4e-k. Also, in fig 6, why is the mutant referred to as CM in some panels and MUT in others?

Finally, the paper is written extremely poorly and needs extensive proof reading for grammar and content.

Other comments:

In the first paragraph of the intro, the authors say that TNBC has worst prognosis due to "lack of proper molecular classification". Most would say that the prognosis is due to lack of targeted therapies, in contrast to ER+ or Her2 overexpressing BrCa.

The intro paragraph about histone modifications is overly generalized. Has it really been rigorously established, for example, that HDAC inhibitors lower recurrence of cancer?

Reviewer #2, Expertise: 2. Breast Cancer (TNBC) and EMT (Remarks to the Author):

In this study, Adhikar et. al. reported that UBR7 is an E3 ligase for H2B monoubiquitination at residue K120. They also found that UBR7 knockdown resulted in certain degrees of changes in H2B120Kub binding to DNA. The authors further showed that UBR7 expression inhibits breast tumor formation and metastasis. They showed that UBR7 loss reduced CDH4 expression and increased Wnt pathway activation, suggesting molecular pathways of UBR7.

UBR7 is a member of the UBR1-7 family of E3 ligase. RNF20/40 (UBR1) was reported as an E3 ligase for H2BK120ub, controlling global transcription changes. UBR7 on the other hand contains a

PHD domain but lacks N-degrons. The major concern for this manuscript is the novelty of the study. What is the relationship between RNF20/40 (UBR1) and UBR7 in H2BK120 monoubiquitination? If both E3 ligases catalyze H2BK120 monoubiquitination, do they share similar profiles on regulating H2BK120ub binding targets and global gene regulation? Current version describes UBR7 as an E3 ligase for H2BK120 monoubiquitination and downstream anti-tumor activity. It is to the reviewer's opinion that this study is suitable to be published in a more specific journal.

Specific comments:

1. The authors used the UBR7 catalytic mutant (UBR7-CM) in assaying for H2BK120 monoubiquitination. In Fig. 1e, 1h, UBR7 CM still showed some activities on H2BK120 monoubiquitination. Is UBR7-CM a dead mutant?
2. Fig. 2g. From the text, it looked like the authors performed ChIP using the H2BK120ub in control and UBR7 KD cells. However, the labels on Fig. 2g suggested that qPCR results were done for UBR7 ChIP validation. If my assumption is correct, the control for Fig. 2g should be SCR shRNA and not IgG.
3. Fig. 2i and Supplementary Fig. 2i. The authors need to clearly describe the approach and also the significance of the three red circles.
4. Fig. 3b. The total numbers of Triple negative breast cancer from TCGA should be much higher than 11. How did the authors select only the 11 cases? ER+/PR+/HER2+ are not commonly compared to Triple negative BC. The authors need to compare non-triple negative and triple negative patients. Also, statistics analysis is required for all comparisons.
5. Fig. 3c and 3d, why the total percentage of red columns adds up to be less than 100% and blue more than 100%?
6. Fig. 4. The authors need to provide tumor cell proliferation data comparing UBR7 overexpression to control tumor cells. It looked like UBR7 overexpression drastically inhibited tumor cell growth. If so, the reduced lung metastasis after UBR7 overexpression in MDA-MB cells can be caused by inhibited tumor cell growth rather than their metastasis potential.
7. Fig. 6a. Overlapping gene targets that are bound by H2BK120ub and differentially expressed upon UBR7 KD resulted in 435 common targets. Among these, how many targets showed downregulation or upregulation of H2BK120ub binding in UBR7 KD cells. If UBR7's activity to ubiquitinate H2BK120ub accounts for the differential gene expression in UBR7 KD cells, there should be a large correlation between the binding capacity of H2BK120ub and differential expression of these targeted genes in UBR7 KD cells.
8. Fig. 6C. The authors need to provide WB results on CDH4 for their claim: "CDH4 expression showed consistent patterns to UBR7 in matched 'normal' and 'malignant' breast cancer lines (Fig. 6c)". A previous report (Ref 35 in the manuscript) showed an inconsistent pattern of CHD4 expression in various cancer cell lines.

Reviewer #3, Expertise: Chromatin modifiers (PHD domain)(Remarks to the Author):

The paper by Adhikary et al. reports that the atypical PHD domain of UBR7 acts as an E3 ligase to modify H2B by ubiquitin at K120 and that this activity is critical for its breast cancer tumor

suppressor function. However, there are substantial concerns as detailed below.

First issue concerns *in vitro* ubiquitination (Figs. 1e-h). As described in the Methods, the ubiquitination reactions were carried out in the presence of 50mM EDTA and 50mM DTT. It is striking how the authors could observe ubiquitination under these conditions. EDTA is a potent metal chelating agent and at 50mM, it would abolish E1 activity by chelating MgCl₂ (5mM) and thereby blocking ATP hydrolysis. EDTA at 50mM would also disable UBR7's PHD domain that requires Zn⁺⁺ for structural coordination. In addition, DTT is a redox reagent containing –SH group and at 50mM, it would effectively dissociate either E1-Ub or E2-Ub thiol ester complex, thereby abolishing ubiquitination.

I would hope that “50mM EDTA and 50mM DTT” are typos. If these truly are the conditions used, Figs. 1e-f need to be repeated under appropriate conditions (0.5mM EDTA and 0.5mM DTT).

There are additional problems with the ubiquitination experiments. a) The authors need to provide quantification and inform the percentage of H2B that was utilized for modification. b) Titration experiments should be performed to compare PHD alone and UBR7. c) The lack of activity by PHD on the nucleosome substrate is puzzling (Fig. 1h). Is H2B K120 accessible in the nucleosome structure?

Second issue concerns *in vitro* binding experiments (Figs. 1b-d). Figs. 1b-c should indicate the percentage of input loaded on the gels. Fig. 3d should employ UBR7-CM mutant. Finally, details for binding experiments, including concentration of the input proteins, should be provided in the Methods.

Thirdly, the evidence for the role of UBR7 PHD in H2B K120 ubiquitination (Figs. a-e) needs to be enhanced. The data should be quantified. It appears that in Fig. 2b, the H2B levels in lanes 2 and 3 are lower than that in lane 1. Therefore, it may not be convincing that UBR7 knockdown by shRNA indeed diminishes H2B K120 ubiquitination. In Fig. 2c, it is unclear whether UBR7-WT and UBR7-CM are resistant to UBR7-sh1. If so, please provide sequence details in the Methods. If not, it is unclear how the rescue would work because UBR7-WT and UBR7-CM mRNAs would be sensitive to degradation by UBR7-sh1. In addition, UBR7 should be compared with RNF20/RNF40 (previously identified E3s for H2B mono-ubiquitination) for their relatively contributions to cellular H2B mono-ubiquitination.

Point-by-point Response to Reviewers

We thank the reviewers for their constructive comments which have been instrumental in improving the manuscript overall. We have performed several new experiments and incorporated the same in the modified version to address all the issues raised by both the reviewers.

Reviewer 1

1. A comparison of this PHD domain sequence with other PHD domains, and a prediction of how widespread (or unique) this ub ligase might be. The authors compare the PHD in UBR7 to RING domains found in other UBR proteins, but they do not do such comparisons across PHD motifs. How conserved are the 2 critical histidine residues?

Answer: We thank the reviewer for his/her comment. The Supplementary Fig. 1c already had the comparison with UBR7 PHD finger and other PHD fingers which are H3K4Me3 binders. We have additionally done sequence alignment of UBR7 with Unmodified H3 binders.

Supplementary Fig. 1d

```
UBR7-PHD      GLYC-I163CKRPYPDPPEDEIPDEMIQ166VV167CE168DF169HGR170HLGAIPPE-----SGDFQEMV173CQ174AC175MKR
AIRE-PHD      EDECAV163CR-----DGG---ELIC166DC167CPRAFHLA168CL169SPP--L-REIPSGTWR---CSS173CLQA
BHC80-PHD     EDFC163SV164CR-----KSG---QLLM166CD167TSRVYHLD168CLDPP--L-KTIPKGMWI---CPR173Q174DQ
BAZ2A-PHD     KVT163CLV164CR-----KGDNDEFLL166CD167GRGCH168IY169CHRPK--M-EAVPEGDWF---CTV173CLAQ
```

Supplementary Fig 1: Sequence alignment of UBR7-PHD finger with PHD finger containing proteins that binds to unmodified histone H3.

We have observed that although His 163 is conserved, but not His 166 across PHD fingers.

2. Clarification/correction of Fig 2F. As currently drawn, it appears that the shScr control affected H2Bub at 7846 genes, while the UBR7 shRNA affected this modification at ~1100 genes. This appears to be a mistake, as later in the paper the authors overlap the 7800+ genes with changes in expression that occur upon UBR7 knock down. This point is quite important and needs to be clarified.

Answer: We are sorry for the confusion created here. The circles of the Venn Diagram represent total number of H2BK120ub binding sites in cells harbouring Control shRNA or UBR7 shRNA. Overlap is shown as well as unique sites in each cell – either containing control shRNA or UBR7 shRNA. We have labelled it better now to avoid confusion. Other confusion is that these are total number of H2BK120ub binding sites and not genes. Figure legend and labelling has been corrected to indicate the same.

In Fig. 6a, we show the overlap between those H2BK120ub binding sites that are present in Control shRNA containing cells but not in UBR7 shRNA containing cells (effectively to derive those that lose H2BK120Ub upon UBR7 loss) and genes differently deregulated upon UBR7 knock-down. Hence this figure is used to demonstrate which sites in the UBR7 shRNA containing cells are associated with

differentially expressed genes. We have changed the labelling in Fig. 6a, Supplementary Fig. 7a & 7b as well as within the text to better reflect the gains or losses of H2BK120ub binding sites with and without gene expression.

3. Addition of example histone H2Bub ChIP profiles at specific genes in Fig 2 to show the degree of differences observed more clearly. Currently, the only profile shown is for CDH4 in Fig 6, and the changes in H2Bub do not appear to be that striking.

Answer: Indeed, the effect on CDH4 is not the most prominent. However, it is associated with most drastic change in gene expression. It is important to consider that incomplete knockdown of UBR7 likely contributes to modest change seen on CDH4 gene. Nonetheless, the biology of CDH4 and functional experiments shown in Fig. 6 and 7 clearly demonstrate its epistatic roles with UBR7. We now show examples of other genes that show more prominent changes in H2BK120ub upon UBR7 knockdown. CDH13 is one such example, however, the gene expression change on this gene was modest. Please find examples of other genes that show differences in Supplementary Fig. 7e and Supplementary Fig. 3k-m. Supplementary Fig. 7c-d contrast changes in CDH4 and CDH13 with neighbouring region.

4. Addition of experiments with the UBR CM construct to the experiments shown in Fig3 K and I, and also Fig 4e-k. Also, in fig 6, why is the mutant referred to as CM in some panels and MUT in others?

Answer: We thank the reviewer for his/her critical comment. We have performed all the experiments with the UBR7 catalytic mutant (CM) construct as done with the UBR7 wild type (WT) constructs as represented in 3k,l, 4h-k. Interestingly we have observed a tumor suppressor phenotype of UBR7-WT protein but not the UBR7-CM in local tumors through mammary fat pad injections (Fig. 4e, f) as well as lung metastasis through tail vein injections (Fig. 4g, h). The phenotype with UBR7-CM was already tested in invasion assays (Fig. 4l,m) and migration assays (Fig. 4n, o). We have also changed the MUT to CM for uniformity in Fig.6.

Figure 3

Figure 3: (k) Proliferation of cultured MCF10A UBR7-sh1 cells expressing vector (VECTOR), wild-type (UBR7 WT) or catalytic mutant (UBR7-CM). (l) qRT-PCR analysis of Ki67 in MCF10A UBR7-sh1 cells expressing vector (VECTOR), wild type (UBR7 WT) or catalytic mutant (UBR7-CM) in a time-dependent manner.

Figure 4

Figure 4: (e-f) Tumor formation in mice injected (fat pad) with MDA-MB-231 cells expressing empty vector (VECTOR) or wild-type (UBR7 WT) or catalytic mutant (UBR7-CM). (g) Tumor metastasis in mice injected (tail vein) with MDA-MB-231 cells expressing empty vector (VECTOR) or wild-type (UBR7 WT) or catalytic mutant (UBR7-CM)

Figure 6

Figure 6: Bar plots showing H2BK120Ub ChIP in CDH4 gene locus in MDA-MB-468 cells expressing wild-type (WT) and catalytic-mutant (CM) UBR7.

5. Finally, the paper is written extremely poorly and needs extensive proof reading for grammar and content.

Answer: We thank the reviewer for pointing this out. As advised, we have done a thorough proof reading of our article. In addition, we got the manuscript proofread by the Scientific Editing team at MD Anderson Cancer Center for grammatical errors.

Other Comments

6. In the first paragraph of the intro, the authors say that TNBC has worst prognosis due to “lack of proper molecular classification”. Most would say that the prognosis is due to lack of targeted therapies, in contrast to ER+ or Her2 overexpressing BrCa.

Answer: We apologize for this misstatement. This has been corrected to “due to lack of targeted therapies” in the revised manuscript.

7. The intro paragraph about histone modifications is overly generalized. Has it really been rigorously established, for example, that HDAC inhibitors lower recurrence of cancer?

Answer: Reviewer is correct in pointing out that some of these statements are likely overstated without definitive proof behind them for ALL cancer types. We have now softened the general statements and pointed to specific cases where the results have been demonstrated by the referred articles. We thank the reviewer for pointing this.

Reviewer 2

1. UBR7 is a member of the UBR1-7 family of E3 ligase. RNF20/40 (UBR1) was reported as an E3 ligase for H2BK120ub, controlling global transcription changes. UBR7 on the other hand contains a PHD domain but lacks N-degrons.What is the relationship between RNF20/40 (UBR1) and UBR7 in H2BK120 monoubiquitination? If both E3 ligases catalyze H2BK120 monoubiquitination, do they share similar profiles on regulating H2BK120ub binding targets and global gene regulation?

Answer: We thank the reviewer for raising this pertinent question. However, it is clear from the amino acid sequence analysis below that RNF20/40 does not belong to the UBR family.

UBR1, RNF20 and RNF40 are constituted of 1749, 975, 1001 amino acids respectively.

UBR1(1749aa) [UniProtKB-Q8I WV7]

10	20	30	40	50
MADEEAGGTE	RMEISAELPQ	TPQRLASWWD	QQVDFYTAFL	HHLAQLVPEI
60	70	80	90	100
YFAEMDPDLE	KQEESVQMSI	FTPLEWYLFQ	EDPDICLEKL	KHSGAFQLCG
110	120	130	140	150
RVFKSGETTY	SCRDCAIDPT	CVLCMDCFQD	SVHKNHRYKM	HTSTGGGFCD
160	170	180	190	200
CGDTEAWKTG	PFCVNHEPGR	AGTIKENSRC	PLNEEVIVQA	RKIFPSVIKY
210	220	230	240	250
VVEMTIWEEE	KELPPELQIR	EKNERYCYVL	FNDEHHSYDH	VIYSLQRALD
260	270	280	290	300
CELAEALQHT	TAIDKEGRRA	VKAGAYAACQ	EAKEDIKSHS	ENVSQHPLHV
310	320	330	340	350
EVLHSEIMAH	QKFALRLGSW	MNKIMSYSDD	FRQIFCQACL	REEPDSENPC

360	370	380	390	400
LISRLMLWDA	KLYKGARKIL	HELIFSSFFM	EMEYKKLFAM	EFVKYYKQLQ
410	420	430	440	450
KEYISDDHDR	SISITALSVQ	MFTVPTLARH	LIEEQNVISV	ITETLLEVLPL
460	470	480	490	500
EYLDNRNKFN	FQGYSDKLG	RVYAVICDLK	YILISKPTIW	TERLRMQFLE
510	520	530	540	550
GFRSFLKILT	CMQGMEEIRR	QVGQHIEVDP	DWEAAIAIQM	QLKNILLMFQ
560	570	580	590	600
EWACACDEELL	LVAYKECHKA	VMRCSTSFIS	SSKTVVQSCG	HSLETKSYRV
610	620	630	640	650
SEDLVSIHLP	LSRTLGLHV	RLSRLGAVSR	LHEFVSFEDF	QVEVLVEYPL
660	670	680	690	700
RCLVLVAQVV	AEMWRRNGLS	LISQVFYYQD	VKCREEMYDK	DIIMLQIGAS
710	720	730	740	750
LMDPNKFLLL	VLQRYELAEA	FNKTISTKDQ	DLIKQYNLI	EEMLQVLIYI
760	770	780	790	800
VGERYVPGVG	NVTKEEVTMR	EIIHLLCIEP	MPHSAIAKNL	PENENNETGL
810	820	830	840	850
ENVINKVATF	KKPGVSGHGV	YELKDESLKD	FNMYFYHYSK	TQHSKAEHMQ
860	870	880	890	900
KKRRKQENKD	EALPPPPPE	FCPAFSKVIN	LLNCDIMMYI	LRTVFERAID
910	920	930	940	950
TDSNLWTEGM	LQMAFHILAL	GLLEEKQQLQ	KAPEEEVTFD	FYHKASRLGS
960	970	980	990	1000
SAMNIQMLLE	KLKGIPQLEG	QKDMITWILQ	MFDTVKRLRE	KSCLIVATTS
1010	1020	1030	1040	1050
GSESIKNDI	THDKEKAERK	RKAEAAARLHR	QKIMAQMSAL	QKNFIETHKL
1060	1070	1080	1090	1100
MYDNTSEMPG	KEDSIMIEES	TPAVSDYSRI	ALGPKRGPVS	TEKEVLT CIL
1110	1120	1130	1140	1150
QEQEQVKIENNAMVLSACVQKSTALTQHRGKPIELSGEALDPLFMDPDL				
1160	1170	1180	1190	1200
AYGTYTGSCGHVMHAVQWKYFEAVQLSSQQRHVDLDFLESGEYLCPLC				
1210	1220	1230	1240	1250
KSLCNTVIPI	IPLQPQKINS	ENADALAQLL	TLARWIQTVL	ARISGYNIRH
1260	1270	1280	1290	1300
AKGENPIPIF	FNQGMGDSTL	EFHSILSFGV	ESSIKYSNSI	KEMVILFATT
1310	1320	1330	1340	1350
IYRIGLKVPP	DERDPRVPM	TWSTCAFTIQ	AIENLLGDEG	KPLFGALQNR
1360	1370	1380	1390	1400
QHNGLKALMQ	FAVAQRITCP	QVLIQKHLVR	LLSVVLPNIK	SEDTPCLLSI
1410	1420	1430	1440	1450
DLFHVLVGAV	LAFPSLYWDD	PVDLQPSSVS	SSYNHLYLFH	LITMAHMLQI
1460	1470	1480	1490	1500
LLTVDTGLPL	AQVQEDSEEA	HSASSFFAEI	SQYTSGSIGC	DIPGWYLWVS
1510	1520	1530	1540	1550
LKNGITPYLR	CAALFFHYLL	GVTPEELHT	NSAEGEYSAL	CSYLSLPTNL
1560	1570	1580	1590	1600
FLLFQYWDT	VRPLLQRWCA	DPALLNCLKQ	KNTVVRYPRK	RNSLIELPDD
1610	1620	1630	1640	1650
YSCLLNQASH	FRCPRSADDE	RKHPVLCLFC	GAILCSQNIC	CQEIVNGEEV
1660	1670	1680	1690	1700
GACIFHALHC	GAGVCIFLKI	RECRVVLVEG	KARGCAYPAP	YLDEYGETDP
1710	1720	1730	1740	
GLKRGNPLHL	SRERYRKLHL	VWQQHCIEE	IARSQETNQM	LFGFNWQLL

RNF20(975aa) [UniProtKB-Q5VTR2]

10	20	30	40	50
MSGIGNKRAA	GEPGTSMPPE	KKAAVEDSGT	TVETIKLGGV	SSTEELDIRT
60	70	80	90	100
LQTKNRKLAE	MLDQRQAIED	ELREHIEKLE	RRQATDDASL	LIVNRYWSQF
110	120	130	140	150
DENIRIILKR	YDLEQQLGDL	LTERKALVVP	EPEPDSDSNQ	ERKDDRERGE
160	170	180	190	200
GQEPAFSFLA	TLASSSSEEM	ESQLQERVES	SRRAVSQIVT	VYDKLQEKVE
210	220	230	240	250
LLSRKLNSGD	NLIVEEAVQE	LNSFLAQENM	RLQELTDLLQ	EKHRTMSQEF
260	270	280	290	300
SKLQSKVETA	ESRVSVLESM	IDDLQWDIDK	IRKREQRLNR	HLAEVLERVN
310	320	330	340	350
SKGYKVYAG	SSLYGGTITI	NARKFEEMNA	ELEENKELAQ	NRLCELEKLR
360	370	380	390	400
QDFEEVTQN	EKLKVELRSA	VEQVVKETPE	YRCMQSQFSV	LYNESLQLKA
410	420	430	440	450
HLDEARTLLH	GTRGTHQHQV	ELIERDEVSL	HKCLRTEVIQ	LEDTLAQVRK
460	470	480	490	500
EYEMLRIEFE	QTLAANEQAG	PINREMRHLI	SSLQNHNHQL	KGEVLRKRYK
510	520	530	540	550
LREAQSDLNK	TRLRSGSALL	QSQSSTEDPK	DEPAELKPDS	EDLSSQSSAS
560	570	580	590	600
KASQEDANEI	KSKRDEEERE	RERREKERER	EREREKEKER	EREKQKLKES
610	620	630	640	650
EKERDSAKDK	EKGKHDDGRK	KEAEI IKQLK	IELKKAQESQ	KEMKLLLDY
660	670	680	690	700
RSAPKEQRDK	VQLMAAEKKS	KAELEDLRQR	LKDLEDKEKK	ENKKMADEDA
710	720	730	740	750
LRKIRAVEEQ	IEYLQKKLAM	AKQEEEEALLS	EMDVTGQAFE	DMQEQNIRLM
760	770	780	790	800
QQLREKDDAN	FKLMSERIKS	NQIHKLLKEE	KEELADQVLT	LKTQVDAQLQ
810	820	830	840	850
VVRKLEEKEH	LLQSNIGTGE	KELGLRTQAL	EMNKRKAMEA	AQLADDLKAA
860	870	880	890	900
LELAQKKLHD	FQDEIVENSV	TKEKDMFNFK	RAQEDISRLR	RKLETTKKPD
910	920	930	940	950
NVPKCDEILM	EEIKDYKARL	TCPCCNMRKKDAVLT	KCFHVFCFECVKTRY	
960	970			
DTRQRKCPKCN	AAFGANDFH	RIYIG		

RNF40(1001aa) [UniProtKB-075150]

10	20	30	40	50
MSGPGNKRAA	GDGGSGPPEK	KLSREEKTTT	TLIEPIRLGG	ISSTEEMDLK
60	70	80	90	100
VLQFKNKKLA	ERLEQRQACE	DELRERIEKL	EKRQATDDAT	LLIVNRYWAQ
110	120	130	140	150
LDETVEALLR	CHESQGELSS	APEAPGTQEG	PTCDGTPLPE	PGTSELRDPL
160	170	180	190	200
LMQLRPPLSE	PALAFVVALG	ASSSEEVELE	LQGRMEFSKA	AVSRVVEASD
210	220	230	240	250
RLQRRVEELC	QRVYSRGDSE	PLSEAAQAHT	RELGRENRRL	QDLATQLQEK
260	270	280	290	300
HHRISLEYSE	LQDKV TSAET	KVLEMETTVE	DLQWDIEKLR	KREQKLNKHL
310	320	330	340	350
AEALEQLNSG	YYVSGSSSGF	QGGQITLSMQ	KFEMLNAELE	ENQELANSRM
360	370	380	390	400
AELEKLQAEI	QGAVRTNERL	KVALRSLPEE	VVRETGEYRM	LQAQFSLLYN
410	420	430	440	450
ESLQVKTQLD	EARGLLLATK	NSHLRHIEHM	ESDELGLQKK	LRTEVIQLED
460	470	480	490	500
TLAQVRKEYE	MLRIEFEQNL	AANEQAGPIN	REMRHLISSL	QNHNHQLKGD
510	520	530	540	550
AQRYKRKLRE	VQAEIGKLRA	QASGSAHSTP	NLGHPEDSGV	SAPAPGKEEG
560	570	580	590	600
GPGPVSTPDN	RKEMAPVPGT	TTTTTSVKKE	ELVPSEEDFQ	GITPGAQGPS
610	620	630	640	650
SRGREPEARP	KRELQEREGP	SLGPPPVASA	LSRADREKAK	VEETKRKESE
660	670	680	690	700
LLKGLRAELK	KAQESQKEMK	LLLDMYKSAP	KEQRDKVQLM	AAERKAKAEV
710	720	730	740	750
DELRSRIREL	EERDRRESKK	IADEDALRRI	RQAEEQIEHL	QRKLGATKQE
760	770	780	790	800
EEALLSEMDV	TGQAFEDMQE	QNGRLLQQLR	EKDDANFKLM	SERIKANQIH
810	820	830	840	850
KLLREEKDEL	GEQVLGLKSQ	VDAQLLTVQK	LEEKERALQG	SLGGVEKELT
860	870	880	890	900
LRSQALELNK	RKAVEAAQLA	EDLKVQLEHV	QTRLREIQPC	LAESRAAREK
910	920	930	940	950
ESFNLKRAQE	DISRLRRKLE	KQRKVEVYAD	ADEILQEEIK	EYKARLT CPC
960	970	980	990	1000
CNTRKKDAVLTKCFHVFCFECVRRGRYEARQRKCPKCN	AAF	GAHDFHRIYI		

UBR7(425aa)[UniProtKB-Q8N806]

10	20	30	40	50
MAGAEGAAGR	QSELEPVVSL	VDVLEEDEEL	ENEACAVLGG	SDSEKCSYSQ
60	70	80	90	100
GSVKRQALYA	CSTCTPEGEE	PAGICLACSY	ECHGSHKLFE	LYTKRNFRC
110	120	130	140	150
CGNSKFKNLE	CKLLPDKAKV	NSGNKYDNF	FGLYCICKRPPDPEDEIPD	
160	170	180	190	200
EMIQCVCEDWFHGRHLGAI	PPESGDFQEMVQACMKR	CS	FLWAYAAQLA	
210	220	230	240	250
VTKISTEDDG	LVRNIDGIGD	QEVIKPENGE	HQDSTLKEDV	PEQGKDDVRE
260	270	280	290	300
VKVEQNSEPC	AGSSSESDLQ	TVFKNESLNA	ESKSGCKLQE	LKAKQLIKKD
310	320	330	340	350
TATYWPLNWR	SKLCTCQDCM	KMYGDLDFVLF	LTDEYDTVLA	YENKGKIAQA
360	370	380	390	400
TDRSDPLMDT	LSSMNRVQQV	ELICEYNDLK	TELKDYLKRF	ADEGTVVKRE
410	420			
DIQQFFEEFQ	SKRRRRVDGM	QYYCS		

While UBR1, RNF20 and RNF40 harbour RING Finger (highlighted in yellow colour in the sequence) as the catalytic module, UBR7 harbours a PHD finger (highlighted in green colour in the sequence), instead.

To address the question of relationship between UBR7 and RNF20/40 complex, we performed ChIP-Seq experiments for H2BK120ub in MCF10a cells where RNF20 and RNF40 were individually knocked down. The results of this experiment are shown in a new Supplementary Fig. 3. In summary, we find that knockdown of all three enzymes individually affects H2BK120Ub status on >95% sites suggesting that all enzymes are required to maintain H2BK120Ub levels. Two key panels are included below, while rest of the results are shown in Supplementary Fig. 3.

h Venn diagram showing overlap of total H2BK120ub binding sites in merged Control (SCR) shRNA expressing MCF10A cells that are lost upon knockdown of UBR7, RNF20 or RNF40. *i* Average genebody density plot for H2BK120ub binding sites in merged Control (SCR) and UBR7, RNF20 or RNF40 shRNA expressing MCF10A cells.

This observation raises several questions that will require further investigation:

1. Does UBR7 physically interact with RNF20 and/or RNF40? RNF20 and RNF40 are known to act as heterodimers.
2. Do UBR7, RNF20 and RNF40 target same loci?
3. Are there distinct roles for UBR7 and RNF20/40 during initiation versus maintenance of this mark?
4. Do UBR7 and RNF20/40 stimulate each other's enzymatic activity?
5. Do UBR7 and RNF20/40 cooperate during spreading of the mark?

These questions will require several biochemical, genomic and functional experiments which we will follow up in coming months. We would like to report all the results together in another manuscript that focuses on deep characterization of relationship between UBR7 and RNF20/40. Hence, we request the reviewer to allow us to exclude the current ChIP-Seq data for H2BK120Ub in RNF20/40 knockdown cells from the final manuscript (Supplementary Fig. 3).

Specific Comments

1. The authors used the UBR7 catalytic mutant (UBR7-CM) in assaying for H2BK120 monoubiquitination. In Fig. 1e, 1h, UBR7 CM still showed some activities on H2BK120 monoubiquitination. Is UBR7-CM a dead mutant?

Answer: We thank the reviewer for raising this issue. We have repeated Fig. 1e (modified Fig. 1f), 1f (modified Supplementary Fig. 1j), 1h (modified Fig. 1g) panels again. Additionally we have performed titration experiment with increasing molar concentration of the E3s UBR7-WT, UBR7-CM and isolated PHD finger with each of the substrates (H2B, H2A/H2B dimer, Core Histones, Nucleosomes) with (Fig. 1h-m, Supplementary Fig. 1 l-o). Cumulatively, our results clearly indicate that UBR7-WT has the best catalytic activity on Nucleosome substrate, followed by core histones, H2A/H2B dimer and free H2B, unlike UBR7-CM.

Figure 1

Figure 1: (f, g) In-vitro ubiquitination assay with recombinant H2B (f), or purified nucleosomes from HeLa cells (g). (h) Bar plots representing percentage of H2BK120Ub formation from H2B using UBR7-PHD, UBR7-WT or UBR7-CM. (i) Activity plot representing the efficiency of wild-type UBR7-WT at optimal concentration, to ubiquitinate histone H2B at K120 at different level of organization. (j, m) Activity plot representing the efficiency of UBR7-PHD, UBR7-WT or UBR7-CM to ubiquitinate histone H2B at K120 at different level of organization, recombinant H2B (j), H2A/H2B dimer (k), core octamer (l) or purified nucleosome from HeLa cells (m).

Supplementary Figure 1

Supplementary Figure 1: (j) In-vitro ubiquitination assay with H2A/H2B dimer. (i-o) In-vitro ubiquitination assay using increasing concentration of UBR7-PHD, UBR7-WT or UBR7-CM with recombinant H2B (i), or H2A/H2B dimer (m), or core octamer (n), or purified nucleosomes from HeLa cells (o).

2. Fig. 2g. From the text, it looked like the authors performed ChIP using the H2BK120ub in control and UBR7 KD cells. However, the labels on Fig. 2g suggested that qPCR results were done for UBR7 ChIP validation. If my assumption is correct, the control for Fig. 2g should be SCR shRNA and not IgG.

Answer: We thank the reviewer for his comment. Fig. 2g shows the occupancy of UBR7 in selective gene targets that were obtained through ChIP-Seq experiments. Hence IgG has been used as negative control. The effect of UBR7 knock down on enrichment of H2BK120Ub on these gene sets were subsequently tested, which clearly indicated that UBR7 knockdown significantly reduces H2BK120Ub levels of chromatin as compared to the scrambled control (Fig. 2i).

3. Fig. 2i and Supplementary Fig. 2i. The authors need to clearly describe the approach and also the significance of the three red circles.

Answer: More detailed information regarding chromatin state analysis has been added to the text and figure legend Fig. 2i as below.

To analyze the impact of UBR7 mediated H2BK120Ub on the chromatin landscape, we also performed ChIP-seq for histone modifications H3K79Me2 (transcription), H3K4Me3 (promoters), H3K4Me1 (enhancers), H3K27Ac (active), H3K27Me3 (polycomb-repressed) and H3K9Me3 (heterochromatin)³¹. Consistent with prior reports, we noted a loss of active transcription mark H3K79Me2 on H2BK120Ub gene targets upon UBR7 knockdown (Fig. 2j). Next, we used the ChromHMM algorithm to determine if any combinatorial chromatin states were influenced by UBR7 depletion. ChromHMM further revealed H2BK120ub was primarily associated with high and low levels of H3K79me2, with the most significant transitions occurring from highly transcribed chromatin states in SCR control cells to low/non-transcribed states in UBR7 depleted cells, including losses of H2BK120ub/K79low (State 1 to 5), H2BK120ub/K79high (State 2 to 1 or 3) and H3K79me2 only (State 3 to 5) (Fig. 2k-l).

Overall, these experiments identify UBR7 as an E3 ubiquitin ligase in vivo and demonstrate importance of UBR7 in maintaining specific chromatin patterns in the cell.

(I) Overlap enrichment analysis displaying chromatin state transitions between MCF10A SCR control cells (Y-axis) and MCF10A UBR7-sh1 cells (x-axis). The most significant state transitions include losses of H2BK120ub/K79 low (State 1 to 5), H2BK120ub/K79 high (State 2 to 1 or 3) and H3K79me2 only (State 3 to 5) which are highlighted by red circles. P-values were calculated using two-tailed t-tests. *P<0.05, **P<0.001 and ***P<0.0001.

Chromatin State Calls:

ChromHMM⁶² was used to identify combinatorial chromatin state patterns based on the histone modifications studied. Normalized bam files were converted into binarized data at a 1000bp resolution using the BinarizeBam command with a p-value cut-off of 1e-5. We specified that ChromHMM should learn a model based on 10 chromatin states. As we considered models between 8 and 20 chromatin states, we chose a 10-state model because it is large enough to identify important functional elements while still being small enough to interpret easily. Overlap Enrichment was used to compute differential enrichment in each of the 10-chromatin states between Control and UBR7-shRNA samples. The ChromHMM segment files from the 10-state model contain the genomic locations of each chromatin state called in both the Control and UBR7-shRNA samples. To determine which chromatin states were enriched between conditions we further compared the genomic locations by using the Control segments file as input for the segment directory, and by further separating the UBR7-shRNA segments file into 10 individual states and using it as input for the external coordinate directory. The UBR7-shRNA segment file was separated into individual chromatin states for the external coordinate directory with the following command:

```
awk -F\t '{print >> $4;close($4)}' ~/path_to/UBR7-sh1_segments.bed
```

Overlap Enrichment was ran using the following command:

```
java -mx4000M -jar ChromHMM.jar OverlapEnrichment ~/path_to/Control_segments.bed  
~/path_to/UBR7-shRNA_segments_separated OverlapEnrichment_Control_vs_UBR7
```

4. Fig. 3b. The total numbers of Triple negative breast cancer from TCGA should be much higher than 11. How did the authors select only the 11 cases? ER+/PR+/HER2+ are not commonly compared to Triple negative BC. The authors need to compare non-triple-negative and triple negative patients. Also, statistics analysis is required for all comparisons.

Answer: We would like to clarify that Fig. 3b is not derived from the TCGA data. It was a representation of the data obtained from tissue microarray. In this data, n was low (n =11) for TNBC cases because a larger number of cores did not stain properly (or were lost during staining) and therefore those datasets were excluded from the analyses. During revision, we stained a new TMA containing a larger number of TNBC cases. This has now allowed us to examine UBR7 levels over 112 TNBC cases in comparison to 57 non-TNBC cases. Please see Fig. 3b (pasted below) for results. In summary, we find that UBR7 staining is significantly reduced in TNBC cases in comparison to non-TNBC (NST) cases. We have also added statistical

analyses to the methods section as well as presented results in the associated Fig. 3b-d.

Figure 3: Bar graph showing percent of samples versus various intensity groups in TNBC vs NST cases.

5. Fig. 3c and 3d, why the total percentage of red columns adds up to be less than 100% and blue more than 100%?

Answer: We apologize for the confusion created here. In Fig. 3c-d, the data was represented in a manner where 100% of samples within a certain intensity group (e.g. 0-1.5) were counted for their ER+ or ER- (or PR + and PR-). We have now replotted the data in another format which represents fraction of ER+ or ER- (or PR + and PR-) samples in the four intensity groups.

6. Fig. 4. The authors need to provide tumor cell proliferation data comparing UBR7 overexpression to control tumor cells. It looked like UBR7 overexpression drastically inhibited tumor cell growth. If so, the reduced lung metastasis after UBR7 overexpression in MDA-MB cells can be caused by inhibited tumor cell growth rather than their metastasis potential.

Answer: We thank the reviewer for the insightful comment. As seen in Fig. 3 and Fig. 4, UBR7 does affect proliferation of the cells. As suggested by the reviewer, newly added Fig. 4g shows difference in Ki-67 staining between control tumors and UBR7 WT overexpressing tumors. Hence it is likely that pro-metastatic effects of UBR7 are in part due to its role in regulating proliferation of cells. Please do note that UBR7 also has drastic effects on invasive ability of the cells (Fig. 4i, j, m, n). To clearly understand the relative contribution of these two effects of UBR7 on metastasis, we need to find a way to uncouple the two activities of UBR7. We currently do not have enough knowledge to devise a way to uncouple these activities (for example, a mutation that could do so). Therefore, we are unable to address this more thoroughly. We have mentioned this possibility in the discussion.

7. Fig. 6a. Overlapping gene targets that are bound by H2BK120ub and differentially expressed upon UBR7 KD resulted in 435 common targets. Among these, how many targets showed downregulation or upregulation of H2BK120ub binding in UBR7 KD cells. If UBR7's activity to ubiquitinate H2BK120ub accounts for the differentially gene expression in UBR7 KD cells, there should be a large correlation between the binding capacity of H2BK120ub and differential expression of these targeted genes in UBR7 KD cells.

Answer: As shown in Fig. 6a, the loss of UBR7 decreased H2BK120ub binding specifically within the promoter (-/+5kbTSS) or gene body of 435 differentially expressed genes (117 up- and 318 down-regulated). As such 435 genes only constitute ~5% of total number of H2BK120Ub sites (~7800) that show loss in UBR7 knockdown cells. However, loss of H2BK120ub may not be sufficient to drive all the steps required to complete gene repression. Furthermore, we are measuring steady-state transcript levels in these RNA-Seq experiments. It is possible that other transient gene repression events are compensated for by other 'non-chromatin' regulatory mechanisms. Lastly, while this was a direct correlation of H2BK120ub binding and gene expression, further investigation is needed to decipher if H2BK120ub may regulate distal genes by regulating DNA looping or 3D chromatin architecture.

8. Fig. 6C. The authors need to provide WB results on CDH4 for their claim: "CDH4 expression showed consistent patterns to UBR7 in matched Normal and malignant breast cancer lines (Fig.6c)". A previous report (Ref 35 in the manuscript) showed an inconsistent pattern of CDH4 expression in various cancer cell lines.

Answer: As advised by the reviewer, we have performed Western Blot analysis with CDH4 antibody. We see a similar trend in RNA and protein level of CDH4 expression. Interestingly, a higher protein level of UBR7 and CDH4 could be seen in normal proliferating (MCF10A/12A) as well as luminal breast cancer cell (MCF7). There is a reduced trend of expression of UBR7 and CDH4 between T47D, MDA-MB-231 and MDA-MB-468 respectively. This clearly indicates UBR7 and CDH4 follows a similar expression pattern, which is peaked at normal proliferating cells, followed by a moderate expression in luminal type of breast cancer and reduced significantly in triple negative breast cancer.

Figure 6: Immunoblot showing the expression of RCAD/CDH4 across different breast normal and cancer cell lines. Densitometric values are given at the bottom of the blot. RCAD was normalized to GAPDH.

Reviewer 3

Major Concerns:

1. First issue concerns in vitro ubiquitination (Figs. 1e-h). As described in the Methods, the ubiquitination reactions were carried out in the presence of 50mM EDTA and 50mM DTT. It is striking how the authors could observe ubiquitination under these conditions. EDTA is a potent metal chelating agent and at 50mM, it would abolish E1 activity by chelating MgCl₂ (5mM) and thereby blocking ATP hydrolysis. EDTA at 50mM would also disable UBR7's PHD domain that requires Zn⁺⁺ for structural coordination. In addition, DTT is a redox reagent containing SH group and at 50mM, it would effectively dissociate either E1-Ub or E2-Ub thiol ester complex, thereby abolishing ubiquitination.

I would hope that "50mM EDTA and 50mM DTT" are typos. If these truly are the conditions used, Figs. 1e-f need to be repeated under appropriate conditions (0.5mM EDTA and 0.5mM DTT).

Answer: At the onset we thank the reviewer for pointing this error in the Materials and Methods section of the manuscript. The 50mM EDTA and 50mM DTT were the stock concentrations used in the assay. The final concentrations turned out to be 5.0 mM EDTA (for negative control) and 1.0 mM DTT which has been elaborated in the revised Methods adhering to the user manual guidelines of the kit (Catalogue No.:BML-UW9920). Similar conditions have been employed in several standard references (*Hepatocyte TRAF3 promotes liver steatosis and systemic insulin resistance through targeting TAK1-dependent signalling: P.X. Wang, et al.; Nat. Commun. 7, 10592 (2016)*).

2. There are additional problems with the ubiquitination experiments. a) The authors need to provide quantification and inform the percentage of H2B that was utilized for modification. b) Titration experiments should be performed to compare PHD alone and UBR7. c) The lack of activity by PHD on the nucleosome substrate is puzzling (Fig. 1h). Is H2B K120 accessible in the nucleosome structure?

Answer: We appreciate the reviewers concern here. To address them we have performed the following additional experiments:

a) The blots have been quantified for H2BK120Ub percent after normalizing to total histone H2B taken (Fig. 1h; the blots used for the calculation are Fig. 1f for H2B, Supplementary Fig. 1j for H2A/H2B dimer, Supplementary Fig. 1k for core histones and Fig. 1g for Nucleosomes).

b) Titration experiments have been performed with increasing molar concentration of UBR7-WT, UBR7-CM or UBR7-PHD finger with of each of the substrates (H2B, H2A/H2B dimer, Core Histones, Nucleosomes) (Fig. 1j-m, Supplementary Fig. 1 l-o). Cumulatively, our results clearly indicate that UBR7-WT has the best catalytic activity on Nucleosome substrate, followed by core histones, H2A/H2B dimer and free H2B. The isolated PHD finger also shows a similar trend with compromised activity as compared to the UBR7-WT. UBR7-CM has significantly compromised catalytic activity.

c) We thank the reviewer again for intuitively raising the question with nucleosome substrate. After carefully revisiting this, we did observe a catalytic activity of the isolated PHD finger with nucleosome substrate (Fig. 1g and Supplementary Fig. 1o).

We have also plotted the nucleosome surface and observed that H2BK120 is surface exposed and quite accessible (Fig. 1n).

Figure 1

Figure 1: (f, g) In-vitro ubiquitination assay with recombinant H2B (f), or purified nucleosomes from HeLa cells (g). (h) Bar plots representing percentage of H2BK120Ub formation from H2B using UBR7-PHD, UBR7-WT or UBR7-CM. (i) Activity plot representing the efficiency of wild-type UBR7-WT at optimal concentration, to ubiquitinate histone H2B at K120 at different level of organization. (j, m) Activity plot representing the efficiency of UBR7-PHD, UBR7-WT or UBR7-CM to ubiquitinate histone H2B at K120 at different level of organization, recombinant H2B (j), H2A/H2B dimer (k), core octamer (l) or purified nucleosome from HeLa cells (m). (n) Nucleosome surface diagram representing lysine120 (marked in red) of histone H2B.

Supplementary Figure 1

Supplementary Figure 1: (j) In-vitro ubiquitination assay with H2A/H2B dimer. (i-o) In-vitro ubiquitination assay using increasing concentration of UBR7-PHD, UBR7-WT or UBR7-CM with recombinant H2B (i), or H2A/H2B dimer (m) core octamer (n), or purified nucleosomes from HeLa cells (o).

3. Second issue concerns in vitro binding experiments (Figs. 1b-d). Figs. 1b-c should indicate the percentage of input loaded on the gels. Fig. 3d should employ UBR7-CM mutant. Finally, details for binding experiments, including concentration of the input proteins, should be provided in the Methods.

Answer: We thank the reviewer for rightly pointing out this omission. We have now included percentage of Input loaded on the gels (10%) from Figs. 1b-e. We have repeated the histone interaction with UBR7-CM and represented it in Fig. 1e. We have also elaborated the Method section for the binding experiments (and included the concentration of the Input proteins) in further details.

Figure 1: (b) Immunoblots showing in-vitro interaction of purified UBR7 with recombinant histones H3, H4, H2B or H2A. (c) Interaction of UBR7 full-length wild-type (WT) or catalytic-mutant (CM) with recombinant H2B, H2A/H2B dimer, core octamer or purified nucleosomes from HeLa cells. (d) Ex-vivo

interaction of H2B with UBR7 wild-type (WT) or mutant (CM) in HEK293T cells. (e) Ex-vivo interaction of UBR7 with H2B wild-type (WT) or H2B mutant (K120R) in HEK293T cells.

4. Thirdly, the evidence for the role of UBR7 PHD in H2B K120 ubiquitination (Figs.a-e) needs to be enhanced. The data should be quantified. It appears that in Fig.2b, the H2B levels in lanes 2 and 3 are lower than that in lane 1. Therefore, it may not be convincing that UBR7 knockdown by shRNA indeed diminishes H2B K120 ubiquitination.

Answer: We thank the reviewer for this important suggestion. In the present Fig. 1 of the manuscript we have tried highlighting the role of PHD finger. To emphasise the role of PHD finger mediating this process, we have done titration experiments with increasing molar concentration of UBR7-WT, UBR7-CM as well as UBR7-PHD, with each of the substrates (H2B, H2A/H2B dimer, Core Histones, Nucleosomes) (Fig. 1j-m, Supplementary Fig. 1 I-o). Our results clearly indicate that the isolated PHD finger also shows a similar trend with reduced activity as compared to the UBR7-WT. We have also quantified %H2BK120Ub formed normalized to the histone H2B used in the reaction (Fig. 1h). We have quantified Fig. 2b-e panels in order to confirm the UBR7 knockdown and its complementation with wild type (WT) and catalytic mutant (CM) proteins indeed alters H2BK120Ub.

Figure 2

Figure: Densitometric values are given at the bottom of the blots. UBR7 was normalized to ACTIN and H2BK120Ub was normalized to H2B.

5. In Fig. 2c, it is unclear whether UBR7-WT and UBR7-CM are resistant to UBR7-sh1. If so, please provide sequence details in the Methods. If not, it is unclear how the rescue would work because UBR7-WT and UBR7-CM mRNAs would be sensitive to degradation by UBR7-sh1.

Answer: We apologize for this unintended omission of shRNA sequences. UBR7 sh1 indeed targets the 3' end of the transcript. The sequence is now included in the methods section.

6. In addition, UBR7 should be compared with RNF20/RNF40 (previously identified E3s for H2B mono-ubiquitination) for their relatively contributions to cellular H2B mono-ubiquitination.

Answer: To address the question of relationship between UBR7 and RNF20/40 complex, we performed ChIP-Seq experiments for H2BK120ub in MCF10a cells where RNF20 and RNF40 were individually knocked down. The results of this experiment are shown in a new Supplementary Fig. 3. In summary, we find that knockdown of all three enzymes individually affects H2BK120Ub status on >95% sites suggesting that all enzymes are required to maintain H2BK120Ub levels. Two key panels are included below, while rest of the results are shown in Supplementary Fig. 3.

h Venn diagram showing overlap of total H2BK120ub binding sites in merged Control (SCR) shRNA expressing MCF10A cells that are lost upon knockdown of UBR7, RNF20 or RNF40. **i** Average genebody density plot for H2BK120ub binding sites in merged Control (SCR) and UBR7, RNF20 or RNF40 shRNA expressing MCF10A cells.

This observation raises several questions that will require further investigation:

1. Does UBR7 physically interact with RNF20 and/or RNF40? RNF20 and RNF40 are known to act as heterodimers.
2. Do UBR7, RNF20 and RNF40 target same loci?
3. Are there distinct roles for UBR7 and RNF20/40 during initiation versus maintenance of this mark?
4. Do UBR7 and RNF20/40 stimulate each other's enzymatic activity?
5. Do UBR7 and RNF20/40 cooperate during spreading of the mark?

These questions will require several biochemical, genomic and functional experiments which we will follow up in coming months. We would like to report all the results together in another manuscript that focuses on deep characterization of

relationship between UBR7 and RNF20/40. Hence, we request the reviewer to allow us to exclude the current ChIP-Seq data for H2BK120Ub in RNF20/40 knockdown cells from the final manuscript (Supplementary Fig. 3).

Reviewers' comments:

Reviewer #2, Expertise: BC and EMT (Remarks to the Author):

The authors have improved the manuscript substantially. This reviewer does not have further comments.

Reviewer #3, Expertise: Ubiquitin ligases and cancer (Remarks to the Author):

The revised paper by Adhikary et al. was improved. However, I still have substantial concerns as detailed below.

Major points:

First, the efficiency with which UBR7 catalyzes the mono-ubiquitination of H2B K120 is not shown. Despite the efforts by the authors to add quantification to their ubiquitination experiments in the revision, they did not calculate the percentage of H2B being modified by ubiquitin in either in vitro assays (Fig. 1) or cell-based experiments (Fig. 2). Judging by Westerns, no detectable slow-migrating forms of H2B, indicative of ubiquitin-modified species, are observed (Figs. 1 and 2). These data suggest extraordinary low efficient H2B ubiquitination reactions driven by UBR7. In contrast, UBR7 binds at least 10% of the input H2B in either free, core octamer or nucleosome form as shown by the pull down experiments (Fig. 1). It is thus puzzling why the ubiquitination efficiency is so low.

Second, in the original critique I asked: "In addition, UBR7 should be compared with RNF20/RNF40 (previously identified E3s for H2B mono-ubiquitination) for their relatively contributions to cellular H2B mono-ubiquitination." Here is the authors' response: "To address the question of relationship between UBR7 and RNF20/40 complex, we performed ChIP-Seq experiments for H2BK120Ub in MCF10a cells where RNF20 and RNF40 were individually knocked down. The results of this experiment are shown in a new Supplementary Fig. 3. In summary, we find that knockdown of all three enzymes individually affects H2BK120Ub status on >95% sites suggesting that all enzymes are required to maintain H2BK120Ub levels. Two key panels are included below, while rest of the results are shown in Supplementary Fig. 3." There is no data for side-by-side comparison between UBR7 and RNF20/RNF40 on the cellular mono-ubiquitination of H2B in the revision. The lack of such data adds to the concerns that UBR7 may not play a significant role in the mono-ubiquitination of H2B K120 in cells.

Minor points:

1) I commented in the original critique that "In Fig. 2c, it is unclear whether UBR7-WT and UBR7-CM are resistant to UBR7-sh1. If so, please provide sequence details in the Methods. If not, it is unclear how the rescue would work because UBR7-WT and UBR7-CM mRNAs would be sensitive to degradation by UBR7-sh1." The authors responded: "We apologize for this unintended omission of shRNA sequences. UBR7 sh1 indeed targets the 3' end of the transcript. The sequence is now included in the methods section." If the authors intended to say that UBR7-sh1 targets the 3' end of mRNA that is non-translated and thus recombinant UBR7-WT and UBR7-CM are resistant to UBR7-sh1, please clarify in the methods section.

2) Fig. 1h: It is unclear what "% of H2BK120Ub" means.

Overall comment: While I do believe that UBR7 can catalyze the mono-ubiquitination of H2B at K120 to some extent, this activity appears to be very low given the data shown in vitro (Fig. 1)

and in cell-based assays (Fig. 2). There are substantial concerns on whether UBR7 indeed acts as E3 for modifying H2B at significant levels. The paper provides no compelling evidence that diminish such concerns.

Reviewer #3:

Major points:

First, the efficiency with which UBR7 catalyzes the mono-ubiquitination of H2B K120 is not shown. Despite the efforts by the authors to add **quantification to their ubiquitination experiments** in the revision, **they did not calculate the percentage of H2B being modified by ubiquitin in either in vitro assays (Fig. 1) or cell-based experiments (Fig. 2).**

Response: We apologize for the same. In all our in-vitro experiments, we have now calculated the levels of H2BK120Ub, by quantifying the band intensity of H2BK120Ub and normalizing it to the total H2B levels. The percentage of H2BK120Ub indicates the percentage of H2B being modified by ubiquitin. Please see (Fig. 1a) which is also pasted below for reviewer's convenience.

In cell based experiments, we have quantified the efficiency of knockdown of UBR7 and the loss of global H2BK120Ub level. We have also plotted the same in a bar graph format. Similarly we have also quantified the formation of H2BK120Ub upon UBR7-WT and UBR7-CM overexpression in MCF10A-sh1, MDA-MB-231 or MDA-MB-468, and represented in bar graphs. Please see (Fig. 1b-e) which is also pasted below for reviewer's convenience.

Figure 1: Percentage of H2B being modified represented as percentage of H2BK120Ub formed either by UBR7-PHD, UBR7-WT or UBR7-CM (a). Bar graphs at the bottom of the blots indicating the level of H2BK120Ub, normalized to H2B (b-e).

Judging by Westerns, **no detectable slow-migrating forms of H2B**, indicative of ubiquitin-modified species, are observed (Figs. 1 and 2). These data suggest **extraordinary low efficient H2B ubiquitination reactions driven by UBR7**. In contrast, UBR7 binds to at least 10% of the input H2B in either free, core octamer or nucleosome form as shown by the pull down experiments (Fig. 1). It is thus puzzling why the ubiquitination efficiency is so low.

Response: We thank the reviewer for this important comment. We tested the total H2B antibody (Cat No.: ab1790 from Abcam) in Input and the immunoprecipitated whole cell extract. We were able to detect the slower migrating band, which corresponds to ~25 kDa H2BK120Ub, only in the immunoprecipitate. We show this in Figure 2 below, where we performed immunoprecipitation of histone H2B from UBR7, RNF20 and RNF40 knockdown cells. We could find a slow migrating band of H2B at ~25kDa, in control lane, which showed significant reduction in the UBR7, RNF20 and RNF40 knockdown lanes. Therefore judging by the band position at 25kDa, indicates that slow migrating form is ubiquitinated H2B which is lost upon the knockdown of the E3 ligases.

The H2BK120Ub antibody detects only the monoubiquitinated H2B. In this experiment, we immunoblotted with H2BK120Ub and H2B antibody separately and aligned them with the molecular weight marker, showing that H2BK120Ub picks up a single band at 25kDa whereas the H2B antibody picks up only one single band at 15kDa (Figure 3). Even with long exposure times we did not detect any slower migrating band of modified H2B using this H2B antibody.

We also tried another approach to address this concern of the reviewer. We performed *in-vitro* ubiquitination assay with different histones as substrate and blotted with anti-Ubiquitin antibody. As expected we could see a higher molecular weight band of ubiquitin only in the histone H2B lane, indicating formation of higher molecular mass of histone H2B, thereby showing a slow-migrating species of H2B in immunoblot (Figure 4a).

We also performed cell based experiment, where we co-expressed FLAG H2B wild-type (WT) and EGFP UBR7. Again we could see a shifted band of FLAG in UBR7 overexpressed cells, indicating higher molecular species of H2B (Figure 4b).

Together, our results indicate efficient ubiquitinating ability of UBR7 both *in-vitro* as well *in-vivo*.

Figure 2: Immunoprecipitation of total H2B from whole cell lysate of Control, UBR7, RNF20 or RNF40 knockdown cells and immunoblotted with H2B antibody.

Figure 3: Immunoblot with H2BK120Ub and H2B either from acid extracted histones from MCF10A cells (a) or from MCF10A whole cell lysate (b).

Figure 4: In-vitro ubiquitination assay with histone H3, H4, H2A and H2B as substrate and UBR7-WT as enzyme (a). Immunoblot of either FLAG H2B WT or FLAG H2B K120R with EGFP UBR7 coexpression.

Thus our results indicate efficient ubiquitinating ability of UBR7 both *in-vitro* as well *in-vivo*.

Second, in the original critique I asked: “In addition, UBR7 should be compared with RNF20/RNF40 (previously identified E3s for H2B mono-ubiquitination) for their relatively contributions to cellular H2B mono-ubiquitination.”,

Here is the authors’ response: “To address the question of relationship between UBR7 and RNF20/40 complex, we performed ChIP-seq experiments for H2BK120ub in MCF10A cells where RNF20 and RNF40 were individually knocked down. The results of this experiment are shown in a new Supplementary Fig. 3. In summary, we find that knockdown of all three enzymes individually affects H2BK120Ub status on >95% sites suggesting that all enzymes are required to maintain H2BK120Ub levels. Two key panels are included below, while rest of the results are shown in Supplementary Fig. 3D”

There is no data for side-by-side comparison between UBR7 and RNF20/RNF40 on the cellular mono-ubiquitination of H2B in the revision. The lack of such data adds to the concerns that UBR7 may not play a significant role in the mono-ubiquitination of H2BK120 in cells.

Response: We thank the reviewer for the comment. We have performed immunoblot analysis of H2BK120Ub where extracts from UBR7 and RNF20/40 knockdown cells were run side-

by-side. Indeed we can see a global loss of H2BK120Ub almost to the same extent upon loss of either UBR7 or RNF20 or RNF40.

Figure 5: Immunoblot of H2BK120Ub upon knockdown of either UBR7 or RNF40 or RNF20. Densitometric values of H2BK120Ub have been plotted normalized to H2B.

In addition, provided ChIP-Seq data shows side-by-side comparison between the RNF20, RNF40 and UBR7 knockdown. These data showed major overlaps between H2BK120Ub peaks that are lost upon either of three enzymes: UBR7, RNF20 and RNF40.

Minor points:

1) I commented in the original critique that In Fig. 2c, it is unclear whether UBR7-WT and UBR7-CM are resistant to UBR7-sh1. If so, please provide sequence details in the Methods. If not, it is unclear how the rescue would work because UBR7-WT and UBR7-CM mRNAs would be sensitive to degradation by UBR7-sh1 & 2; The authors responded: “We apologize for this unintended omission of shRNA sequences. UBR7 sh1 indeed targets the 3’UTR; end of the transcript. The sequence is now included in the methods section”; If the authors intended to say that UBR7-sh1 targets the 3’UTR; end of mRNA that is non-translated and thus recombinant UBR7-WT and UBR7-CM are resistant to UBR7-sh1, please clarify in the methods section.

Response: We have clarified in the methods section that UBR7-sh1 indeed targets the 3’UTR, end of mRNA that is non-translated and thus recombinant UBR7-WT and UBR7-CM are resistant to UBR7-sh1.

2) Fig. 1h: It is unclear what does “% of H2BK120Ub” means.

Response: We apologize for not making this clear. Please note that “% of H2BK120Ub” indicates the percentage of H2BK120Ub formed as the product upon catalysis of H2B as the substrate by the enzyme UBR7-PHD or UBR7-WT or UBR7-CM.

Reviewers' comments:

Reviewer #3 (Remarks to the Author):

The revised paper by Adhikary et al. was improved by new data.

However, there appears to be a serious mistake in Fig. 1h, where the authors attempted to quantify the efficiency with which UBR7 catalyzes the mono-ubiquitination of H2B. Based on this graph, H2B in the Core Octamer or Nucleosome is mono-ubiquitinated with very high efficiency at ~55% or ~85%, respectively. This is impossible because as shown in Fig. 1g, where the Nucleosome was used as substrate, the levels of H2B between lane 3 (with wild type UBR7) and lane 4 (with UBR7 CM mutant) remained the same after the reaction. If the mono-ubiquitination efficiency was >80% as claimed by the authors, one would expect to observe very little H2B in lane 3, because a majority of H2B would be converted to the mono-ubiquitinated form, which migrates as a 25KDa band.

I have no idea how the quantification was done. The authors claimed using some "normalization" method without providing details. The authors should have shown the image of anti-H2B Western blots (Figs. 1f and 1g) that include the area from 15 to 25KDa, like the blot shown in Fig. 4b in their rebuttal letter. This would reveal both unmodified and mono-ubiquitinated forms of H2B, the ratio of which could be accurately quantified.

Reviewer #3:

Major points:

First, there appears to be a serious mistake in Fig. 1h, where the authors attempted to quantify the efficiency with which UBR7 catalyzes the mono-ubiquitination of H2B. Based on this graph, H2B in the Core Octamer or Nucleosome is mono-ubiquitinated with very high efficiency at ~55% or ~85%, respectively. This is impossible because as shown in Fig. 1g, where the Nucleosome was used as substrate, the levels of H2B between lane 3 (with wild type UBR7) and lane 4 (with UBR7 CM mutant) remained the same after the reaction. If the mono-ubiquitination efficiency was >80% as claimed by the authors, one would expect to observe very little H2B in lane 3, because a majority of H2B would be converted to the mono-ubiquitinated form, which migrates as a 25KDa band. I have no idea how the quantification was done. The authors claimed using some $\% \text{ normalization}$ method without providing details. The authors should have shown the image of anti-H2B Western blots (Figs. 1f and 1g) that include the area from 15 to 25KDa, like the blot shown in Fig. 4b in their rebuttal letter. This would reveal both unmodified and mono-ubiquitinated forms of H2B, the ratio of which could be accurately quantified.

Response: We apologize for this unintentional mistake. All of the previously shown in-vitro experiments were performed using an anti-H2B antibody (Cat No: ab1790, abcam), which has an epitope at the C-term region. This antibody could not detect both modified and unmodified species of H2B in our in-vitro experiments. During this revision cycle, we were able to identify a different anti-H2B antibody (Cat No. ab18977, abcam) with an epitope in the N-term region, which can easily detect both monoubiquitinated and unmodified forms of H2B. Therefore, with the use of this antibody, we were able to quantitate the reduction of unmodified histone H2B after the ubiquitination reactions. In the updated Figure 1, we show the output of these ubiquitination reactions with either full length UBR7-WT (lane 3) or UBR7-PHD (lane 2) in comparison to UBR7-CM (lane 4) on different substrates including histone H2B (Figure 1a), H2A/H2B dimer (Figure 1b), core octamer (Figure 1c) and nucleosomes (Figure 1d).

Based on these H2B blots, we have calculated the % of H2BK120Ub generated from H2B. Employing ImageJ software, we have quantified the total H2B (both monoubiquitinated and unmodified) and the monoubiquitinated H2B from the same blot using following formula:

$\% \text{ of H2BK120Ub} = (\text{Band Intensity of monoubiquitinated H2B} / \text{Band Intensity of total H2B}) * 100.$

All the calculations are provided in a separate excel sheet (Please see Supplementary Table 7).

We have also shown the image of anti-H2B western blots that includes the area from 15kDa to 25kDa. This has revealed both unmodified and monoubiquitinated forms of H2B, the ratio of which could be accurately quantified.

Figure 1: *In-vitro* ubiquitination assay with different domains of UBR7, UBR7-WT and UBR7-CM as the E3ligase and histone H2B, H2A/H2B dimer, core octamer and nucleosome as the substrate (a-d). Percentage of H2B being modified represented as percentage of H2BK120Ub formed either by UBR7-PHD, UBR7-WT or UBR7-CM (e).